# Antisense oligonucleotide activity in tumour cells is influenced by intracellular LBPA distribution and extracellular vesicle recycling

Alexander N. Kapustin [1✉], Paul Davey[2], David Longmire[2], Carl Matthews [3], Emily Linnane[4], Nitin Rustogi [5], Maria Stavrou[5], Paul W. A. Devine[5], Nicholas J. Bond[5], Lyndsey Hanson[6], Silvia Sonzini [1], Alexey Revenko [7], A. Robert MacLeod[7], Sarah Ross[4], Elisabetta Chiarparin[2] & Sanyogitta Puri[1]

Next generation modified antisense oligonucleotides (ASOs) are commercially approved new therapeutic modalities, yet poor productive uptake and endosomal entrapment in tumour cells limit their broad application. Here we compare intracellular traffic of anti *KRAS* antisense oligonucleotide (AZD4785) in tumour cell lines PC9 and LK2, with good and poor productive uptake, respectively. We find that the majority of AZD4785 is rapidly delivered to CD63+late endosomes (LE) in both cell lines. Importantly, lysobisphosphatidic acid (LBPA) that triggers ASO LE escape is presented in CD63+LE in PC9 but not in LK2 cells. Moreover, both cell lines recycle AZD4785 in extracellular vesicles (EVs); however, AZD4785 quantification by advanced mass spectrometry and proteomic analysis reveals that LK2 recycles more AZD4785 and RNA-binding proteins. Finally, stimulating LBPA intracellular production or blocking EV recycling enhances AZD4785 activity in LK2 but not in PC9 cells thus offering a possible strategy to enhance ASO potency in tumour cells with poor productive uptake of ASOs.

[1] Advanced Drug Delivery, Pharmaceutical Sciences, BioPharmaceuticals R&D, AstraZeneca, Cambridge, UK. [2] Chemistry, Research and Early Development, Oncology R&D, AstraZeneca, Cambridge, UK. [3] Antibody Discovery & Protein Engineering, R&D, AstraZeneca, Cambridge, UK. [4] Bioscience, Research and Early Development, Oncology R&D, AstraZeneca, Cambridge, UK. [5] Analytical Sciences, Biopharmaceutical Development, R&D, AstraZeneca, Cambridge, UK. [6] Bioscience, Research and Early Development, Oncology R&D, AstraZeneca, Alderley Park, UK. [7] Ionis Pharmaceuticals, Carlsbad, CA 92010, USA. ✉email: alexander.kapustin@astrazeneca.com

Antisense oligonucleotides (ASOs) are novel, highly specific nucleic acids designed to silence target genes by forming complimentary heteroduplex with target's mRNA and mediating mRNA degradation[1–3]. Modification of oligonucleotide backbone and sugar moieties has resulted in the development of nuclease-resistant forms which are currently in the pre-clinical or clinical validation for treating a broad range of diseases including SARS-CoV-2, inflammatory diseases, neurological disorders and cancer[1,3–5]. In particular, phosphorothioate ASO (PS-ASO) rapidly redistributes to tissues upon subcutaneous or intravenous injection; yet cytosolic delivery to reach target's mRNA (called productive uptake) remains the most significant challenge[6–8]. Cellular uptake of PS-ASO begins upon binding to cell membrane proteins[9] including scavenger receptors[10], stabilins[11], and EGFR[12], these complexes are endocytosed via clathrin-dependent and independent pathways[9,13] and delivered to early and late endosomes[13,14]. During the final phase, PS-ASO escapes from late endosomes (LE) to the cytosol with the assistance of lysobisphosphatidic acid (LBPA), and annexin A2[14,15]. Several reports have shown that the bulk PS-ASO is entrapped in endosomes and degraded in lysosomes (poor productive uptake) yet the exact intracellular trafficking mechanisms behind the productive uptake remain largely unknown[7–9,13,16,17].

Endocytic pathways merge in LE where internalised cargo can either be sorted for the plasma membrane recycling, loaded to intraluminal vesicles or progress to lysosomal degradation[18,19]. Intraluminal vesicles, in turn, can be secreted as extracellular vesicles (EVs) and these are implicated in the intercellular communication by delivering biologically active proteins, lipids and nucleic acids[19–23]. Intraluminal EV budding in LE is driven by LBPA and Alix[24], and both components are indispensable for ASO endosomal escape most likely acting via the "back-fusion" mechanisms[15]. We and others recently showed that the exogenous siRNA or mRNA delivered in lipid nanoparticle formulation are recycled in EVs[25,26] but it is still unknown whether unformulated ASO could be recycled via this pathway. To understand the role of LE in productive ASO uptake, we investigated intracellular trafficking of next-generation KRAS targeting PS-ASO (AZD4785) with constrained ethyl modifications[4] in two lung tumour cell lines, PC9 with good productive uptake (IC$_{50}$ ≤ 0.6 µM) and LK2 with poor productive uptake (IC$_{50}$ ≥ 10 µM)[17]. We found that bulk of the internalised AZD4785 is delivered to CD63+LE and productive uptake is influenced by LBPA spatial distribution—with LBPA colocalised with CD63+LE in PC9 cells whilst LBPA presented in LEs distinct from CD63+LE across LK2 cells. Notably, AZD4785 is also recycled from CD63+LE in EVs by both cell lines however AZD4785 clearance from LK2 cells via this pathway is greater than in PC9 cells. Moreover, stimulating LBPA production or inhibiting EV recycling pathway by using small molecules enhanced AZD4785 productive uptake in LK2 cells but not in PC9 thus indicating specificity towards poor productive uptake cell lines. Hence, targeting these intracellular pathways could be exploited to boost ASO productive uptake and activity in poor productive tumour cells and, possibly, even across other pathologies.

## Results

**Majority of AZD4785 is delivered to CD63+LE across both cell lines but only limited colocalization with LBPA+LE is observed.** Endosomal ASO escape occurs in the LE[15] so we hypothesised that ASO productive uptake is associated with variable delivery to LE. To test this further, we studied productive uptake of AZD4785[4] (A cEt-modified ASO) in 2 cell lines, PC9 and LK2 which we previously identified to be good and poor productive uptake cells respectively[17]. Cellular AZD4785 uptake was quantified by immunofluorescence using an antibody detecting phosphorothioate backbone[17]. To visualise LE's, we used two LE markers, tetraspanin CD63[27] and LBPA, a lipid which is exclusively generated in LE where it enables ASO endosomal escape by mediating back-fusion of ASO-loaded intraluminal vesicles with the LE membrane[15].

The baseline number of CD63+LEs was lower in LK2 cells, but these were stained more intensively indicating higher CD63 loading per single LE (Fig. 1a–c). In agreement with our previous data[17], there was no difference in bulk AZD4785 uptake by PC9 and LK2 cells (Fig. 1a, d, e). Internalised AZD4785 showed nearly complete overlap with CD63 LEs in both cell lines (Pearson correlation coefficient (PCC) 0.83 and 0.68 ± 0.01 in PC9 and LK2 cells, respectively (Fig. 1f)). By plotting the number of AZD4785 and CD63 spots for individual single cells, we found that the number of intracellular AZD4785 spots, as well as intensity, depends on the CD63+LE level in both cell lines (Fig. 1g, h and Supplementary S1a, b).

LBPA is exclusively presented in LE where it assists ASO escape[15,28,29] so next, we investigated AZD4785 delivery to LBPA-positive LE. LBPA staining revealed distribution across endosome-like spots in the cytosol with the overall intracellular LBPA levels increasing with time (Fig. 2a–c). Next, we investigated whether LBPA is presented in CD63+LE's to the same extent across both cells. As expected, LBPA colocalised with CD63 in PC9 cells (Pearson correlation coefficient (PCC) 0.62 ± 0.14) indicating co-distribution of LBPA and CD63 in the same LE population. Surprisingly, no colocalisation was observed in LK2 cells (PCC = 0.34 ± 0.21) indicating that LBPA is perhaps located in a separate LE population in these cells (Supplementary Fig. S1c, d). Colocalization analysis confirmed AZD4785 delivery to LBPA LE in PC9 cells with PCC reaching 0.54 ± 0.01 after 24 h treatment (Fig. 2d). However, no colocalization between AZD4785 and LBPA was observed in LK2 cells (Fig. 2d). Interestingly, LK2 cells with the highest LBPA content appear to internalise the least levels of AZD4785 (Fig. 2a).

**PS-ASO is recycled in EVs in vitro and in vivo.** Upon delivery to LE, cargo is either sorted for degradation in lysosomes or secreted from cell in EVs[18,19]. Although AZD4785 lysosomal delivery has been extensively studied[3,17] it is currently unknown whether PS-ASO is loaded into EVs. A recent study confirmed a high rate of EV secretion by tumour cells with 2–3 CD63+LEs fusing with the plasma membrane each minute[30] so next we tested whether AZD4785 is recycled via EVs pathway. We incubated both cells with AZD4785 and isolated EVs from conditioned cell culture media using differential ultracentrifugation (see 'Extracellular vesicles isolation and characterisation' section). Nanoparticle Tracking Analysis (NTA) revealed that control, non-treated PC9 and LK2 cells secret EVs having similar average sizes (166.9 ± 5.1 nm and 163.6 ± 4.7 nm, correspondingly) (Fig. 3a, b). Notably, AZD4785 exposure stimulated EVs secretion both by PC9 and LK2 cells (Fig. 3c) and LK2 secreted larger EVs (271.5 ± 12.4 nm) (Fig. 3b). Interestingly, LK2 treatment with PS-ASO and a different sequence (control ASO) also stimulated the secretion of larger EVs (Fig. 2b). Moreover, AZD4785 treatment of other poor productive uptake cells (Calu6 and A427[17]) also resulted in the secretion of larger EVs (Supplementary Fig. S2b) yet good productive uptake cells H460 but not MiaPaca2 also secreted larger EVs (Supplementary Fig. S2a). Wash-out of AZD4785 from good and poor productive uptake cells (pulse-chase) reversed EV production to the control levels (Fig. 3b, c and Supplementary Fig. S2a, b).

Tumour cells secrete heterogenous EVs originating either from LE (exosomes) or plasma membrane (ectosomes)[31,32]. To test

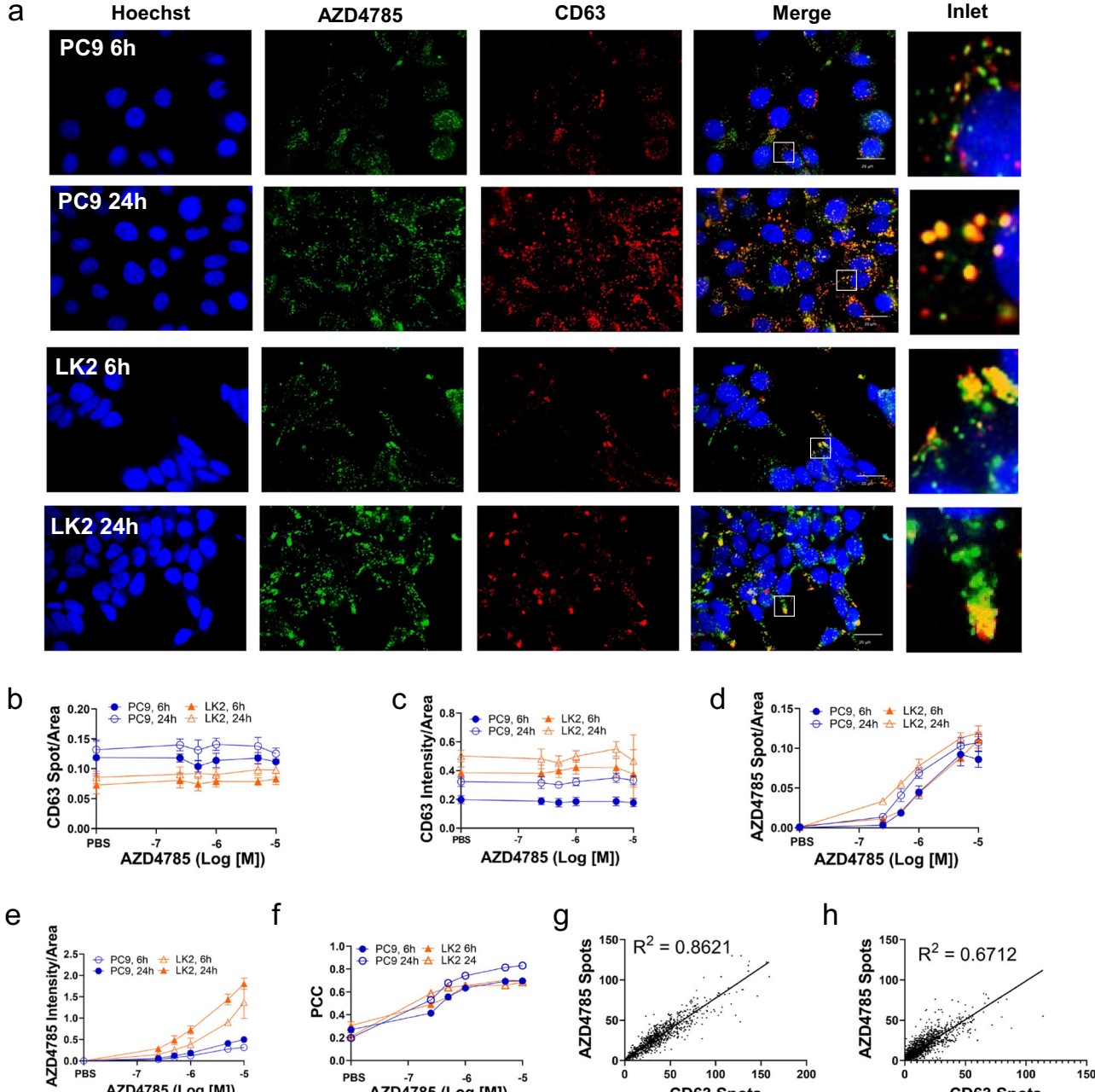

**Fig. 1 AZD4785 is delivered to the CD63-positive LE compartment in PC9 and LK2 cells. a** PC9 and LK2 cells were incubated with AZD4785 for 6 or 24 h, then fixed and stained. LE and AZD4785 were stained with anti-CD63 and anti-PS-ASO antibodies, respectively and visualised using secondary fluorescently labelled antibodies. Nuclei were stained with Hoechst 33342. Cell images were acquired using Opera microscope (×60 objective, NA1.4). **b**–**e** Intracellular content of AZD4785 and CD63 were quantified by using Columbus software by counting number of stained spots in the cell cytosol (Spot/Area) or integrated intensity (Intensity/Area). Number of spots and integrated intensity were normalised to the cellular cytosol area. Average of means, error bars, standard deviation. Minimum 300 cells for each condition were counted in each experiment ($N = 3$–6). **f** AZD4785 and CD63 colocalisation were calculated by Pearson's correlation coefficient (PCC). $N = 3$, error bars, standard deviation (SD). **g** Correlation between intracellular AZD4785 and CD63 spots in single PC9 cells. 24 h treatment with AZD4785. $N = 3$, $n = 948$ cells. Linear regression analysis. **h** Correlation between intracellular ASO and CD63 spots in single LK2 cells. 24 h treatment with AZD4785. $N = 3$, $n = 1759$ cells. Linear regression analysis.

EV's secreted from the LE compartment, we measured the presence of well-established LE markers, CD63 and Alix[31] by western blotting and dot blot. Both markers were presented in PC9 and LK2-derived EVs indicating the presence of EV with LE origin (Fig. 3d and Supplementary Fig. S3a). Interestingly, we also noted lower levels of Alix protein in LK2 cells as compared to PC9 cells (Supplementary Fig. S3b). To test whether AZD4785 is recycled in EVs, we exploited dot blot analysis by using an anti-

phosphorothioate backbone antibody. AZD4785 was detected in EVs isolated from the PS-ASO treated both cell lines and a similar extent of staining was observed in the non-permeabilising and permeabilising conditions indicating that AZD4785 is presented on the outer EV surface (Supplementary Fig. S3a). Albumin has been previously detected in EV's[31] and it is known that over 90% of ASO's with a phosphorothioate backbone are associated with albumin which is responsible for extending ASO plasma

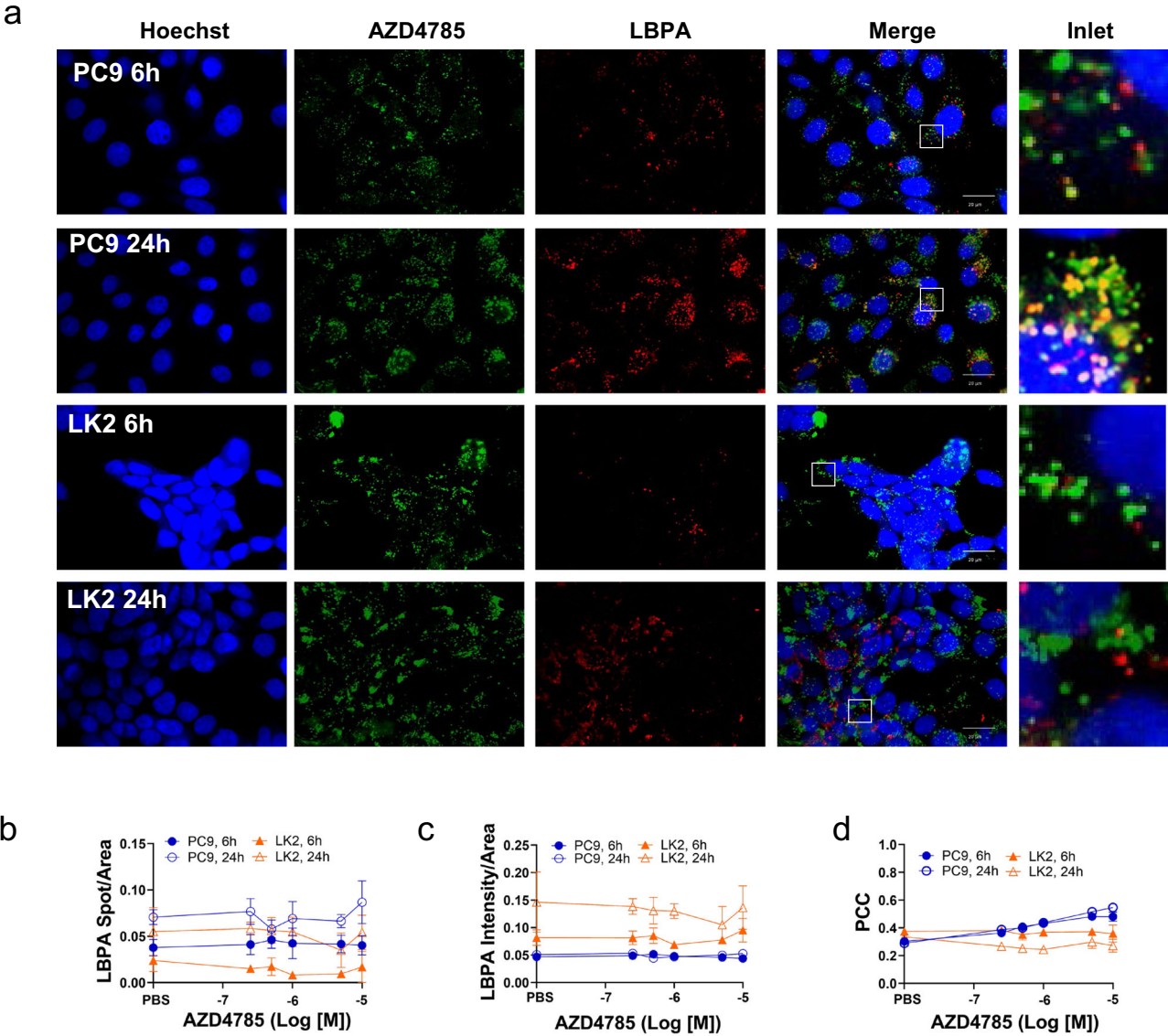

**Fig. 2 AZD4785 is delivered to the LBPA-positive LE compartment in PC9 but not in LK2 cells. a** PC9 and LK2 cells were incubated with AZD4785 for 6 or 24 h, then fixed and stained. LE and AZD4785 were stained with anti-LBPA and anti-PS-ASO antibodies, correspondingly and visualised using secondary fluorescently labelled antibodies. Nuclei were stained with Hoechst 33342. Cell images were acquired using Opera microscope (×60 objective, NA1.4). **b–d** Intracellular content of LBPA were quantified by using Columbus software by counting number of stained spots in the cell cytosol (Spot/Area) or integrated intensity (Intensity/Area). Number of spots and intensity values were normalised to the cellular cytosol area. Average of means. error bars, standard deviation. Minimum 300 cells for each condition were counted in each experiment ($N = 3$–$6$).

circulation time[6,33,34]. Hence, we hypothesised that AZD4785 is delivered to EVs via LE in a complex with albumin[35]. Using isothermal titration calorimetry (ITC), we found that PS-ASO binds to albumin with low affinity (Kd = $8.18 \pm 0.52\,\mu$M, Supplementary Fig. S3c) and this also corroborates well with previous data[33]. Further, we investigated the colocalization of albumin-Alexa594 and PS-ASO in LE across both cells and observed only a small overlap between albumin and AZD4785 in CD63+LEs (Supplementary Fig. S4a). Moreover, albumin endocytosis was significantly suppressed by AZD4785 (Supplementary Fig. S4a–c).

Next, we characterised EV composition further by using proteomic analysis. We identified 1310 proteins in PC9-derived EVs and 952 proteins in LK2-derived EVs (Supplementary Data 1). Both exosome-specific (CD9, CD63, CD81 and synthenin-1) and ectosome-specific (alpha-actinin-4, basignin and F42 cell antigen) markers[31,32] were detected in EV secreted

by both cell lines indicating high EV heterogeneity (Supplementary Data 1). Prominently, 818 proteins were common among all datasets (Fig. 3e). AZD4785 treatment resulted in the appearance of 28 unique proteins in PC9-derived EVs and 53 unique proteins in LK2-derived EVs (Supplementary Data 2). To analyse functional enrichment and interaction between unique protein dataset, we used the STRING database[36]. To differentiate between EVs originating from PC9 and LK2 cells, we included only unique proteins from each dataset. PC9 EVs contained 8 proteins associated with endosomal membrane and intracellular vesicular transport (WLS, SNF8, Rab8B, Rab22A, LAMTOR3, ITM2B and ANTRXR1) and this may reflect their endosomal origin (Supplementary Data 2 and Supplementary Fig. S5a). AZD4785 treatment resulted in the secretion of signalling components (NOTCH3, ERBB2) as well as proteins involved in trans-Golgi network transport vesicles (TGOLN2 and AP1G1) (Supplementary Data 2 and Supplementary Fig. S5b). DNA repair protein

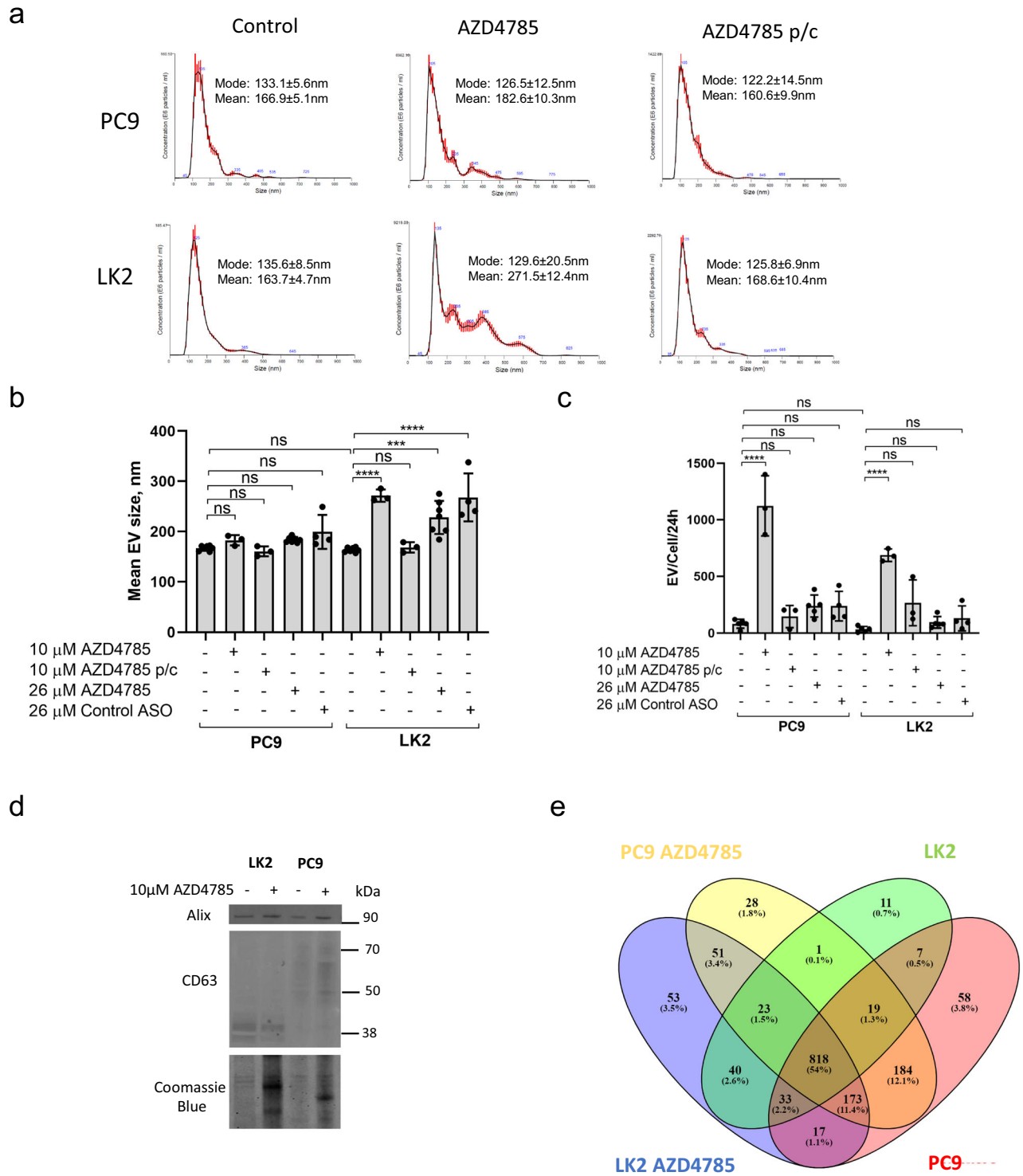

**Fig. 3 AZD4785 stimulates EV secretion by PC9 and LK2 cells. a–c** PC9 and LK2 were treated with AZD4785 for 24 h. For the pulse-chase (p/c) experiment, cells were treated with 10 μM AZD4785 for 6 h, washed and incubated for 24 h in the absence of AZD4785 (AZD4785 p/c). EVs were isolated from conditioned cell media by differential ultracentrifugation and a 100,000×$g$ pellets were analysed by NTA. Representative NTA images. ($N = 6$). **b** Size distribution of EVs secreted by PC9 and LK2 cells. EVs were collected and analysed as in **a**. Error bars, standard deviation. ANOVA ****$p < 0.0001$. ***$p < 0.001$. ns non-significant. $N = 3$–6. **c** Quantification of EV secretion by PC9 and LK2 cells. The total number of EVs was measured by NTA as in **a** and divided by the total number of cells. Error bars, standard deviation. ANOVA. ****$p < 0.0001$, ns non-significant. $N = 3$–6. **d** Detection of exosomal markers, CD63 and Alix, in EVs isolated from PC9 and LK2 cells after 24 h treatment. Equal aliquots of EVs isolated as in **a** were analysed by western blotting and probed for CD63 and Alix. Representative image from $N = 2$. **e** Vein diagram for the EV's protein mass spectrometry analysis revealed differentially secreted protein in PC9 and LK2 EVs.

network including RPA2 and OLB were identified in EVs secreted by LK2 in the control conditions (Supplementary Data 2 and Supplementary Fig. S6a). In addition, the endosomal sorting complex required for transport III component, CHMP2B was detected in LK2 EVs, indicating the presence of LE-derived EVs. AZD4785 treatment changed LK2 EV profile and 15 proteins involved in RNA binding and processing were identified (Supplementary Data 2 and Supplementary Fig. S6b). To the best of our knowledge, these proteins were not previously implicated in PS-ASO intracellular trafficking.

To quantify AZD4785 content in secreted EVs, we applied our recently developed quantitative UPLC-MS assay[17] (Supplementary Figs. S8 and S9). From the standard calibration curve, the sensitivity for intact AZD4785 detection was ≈1 nM (0.005 μg/ml) (Supplementary Fig. S9a). Notably, we found that EVs contained intact AZD4785 and we observed dose-dependent increase in AZD4785 content in EVs secreted by LK2 and PC9 cells (Fig. 4a). Notably, PC9 EVs contained 4.2 fold less AZD4785 then LK2-derived EVs (2103 and 9012 molecules per PC9 or LK2-derived EV at 26.1 μM AZD4785 treatment, respectively) (Fig. 4a). Interestingly, comparison of AZD4785 EV load across the panel of tumour cells with good (PC9, MiaPaca2 and H460) and poor productive uptake cells (LK2, Calu6 and A427) showed the higher AZD4785 loading in EVs produced by non-productive cells (Supplementary Fig. S2c) except H460 which also secreted larger EVs in response to AZD4785 treatment (Figs. S2a, c). Removal of AZD4785 from the conditioned media (pulse-chase) reduced EV AZD4785 load in PC9 cells ≈3 fold times and in LK2 cells ≈6.5 fold times so both cell lines secreted similar amount of AZD4785 in EVs (Fig. 4a). Next, we compared the amount of AZD4785 recycled in EVs with the intracellular AZD4785 content per cell (Fig. 4b). The comparison between EV and intracellular fraction revealed that PC9-derived EVs contain 1/5th (22.8 ± 6.5%) and LK2-derived EVs contain nearly half (44.8 ± 7.5%) of PS-ASO as compared to their respective intracellular uptake levels. Altogether these data indicate that AZD4785 recycling in EVs can represent novel clearance pathway in LK2 cells.

Finally, to test whether PS-ASO recycling via EVs also occurs in vivo, we isolated EVs from plasma of mice with PC9 xenograft tumour after subcutaneous dosing with two AZD4785 doses, 125 mpk/wk and 250 mpk/wk (see 'Mice PC9 xenograft model and EV isolation from plasma' section). NTA analysis showed that the mode size of isolated EVs was 125–130 nm across all conditions (Supplementary Fig. S10a) similar to the in vitro samples. To test whether EVs are secreted by the grafted tumour human PC9 cells, we exploited anti-CD63 antibody that are highly specific to CD63 originating from human (PC9) but not murine cells. We were unable to observe any CD63 specific signal in the plasma-derived EVs as opposed to the CD63 in EVs derived from various human tumour cell lines including PC9. Altogether these data suggest that the majority of isolated EVs are unlikely secreted by xenograft cells and can be secreted by other cells (Supplementary Fig. S10b).

Next, we measured AZD4785 concentration in EVs and original plasma samples (Table 1). AZD4785 concentration 72 h post-dosing in the whole plasma was 12.48 and 18.72 nM for the animals dosed with 125 mpk/wk and 250 mpk/wk AZD4785, correspondingly. In samples post 7 days dosing the AZD4785 plasma concentration reduced to 4.286 nM (125 mpk/wk group) and 6.57 nM (250 mpk/wk group). Next, we quantified AZD4785 content in plasma-derived EVs and found that the 72 h post-dosing EVs contained 0.0706 and 0.0711 fMol of AZD4785 per $10^6$ EVs for the doses 125 mpk/wk and 250 mpk/wk, correspondingly. In the 7 days post-dosing plasma, the concentration of AZD4785 diminished to 0.0305 fMol/$10^6$ EVs and 0.0195 fMol/$10^6$ EVs for doses 125 mpk/wk and 250 mpk/wk,

respectively. Knowing the EV concentration in the plasma, we estimated that ≈5% and ≈2% of plasma PS-ASOs can be transported in EVs, after 72 h and 168 h respectively (Table 1).

**AZD4785 recycling in EVs enables intercellular AZD4785 transfer but reduces productive uptake.** Exosomes efficiently transfer various nucleic acid species between cells[23] so we hypothesised that AZD4785 can be transferred between cells in EVs causing *KRAS* knockdown. We isolated AZD4785-loaded EVs from PC9 and LK2 cells and transferred them to the untreated PC9 and LK2 cells. The exact concentration of AZD4785 delivered via EVs was measured by UPLC-MS as described in the 'Methods' section (see 'AZD4785 UPLC-MS analysis' section) so we directly compared efficacy of carrier-free AZD4785 and AZD4785 delivered in EVs in the concentration range from 3–10 nM for PC9-derived EVs and 10–30 nM for LK2-derived EVs (Fig. 4c–e). Staining of recipient cells with anti-ASO antibody revealed an efficient EV-mediated AZD4785 delivery (Fig. 4c). Moreover, treatment with low doses of EV-associated AZD4785 (10 and 20 nM for PC9 and LK2-derived EVs, respectively), resulted in the intracellular levels similar to those after the treatment with 250 nM carrier-free AZD4785 (Fig. 4c and Supplementary Fig. S7a, b) thus showing that AZD4785 delivery in EV is very efficient.

Next, we tested AZD4785 EV *KRAS* knockdown efficacy. We observed high potency of carrier-free AZD4785 in PC9 cells (Fig. 4d) and low potency in LK2 cells (Fig. 4e) in agreement with the previous data[17]. AZD4785-loaded EVs induced efficient *KRAS* dose-dependent knockdown in PC9 cells with nearly 50% of *KRAS* knockdown even at the low EV-loaded AZD4785 dose (3 nM) (Fig. 4d). Interestingly, there were no significant differences in potency between PC9 and LK2-derived EV (Fig. 4d). Again, the addition of LK2-derived AZD4785 EV to LK2 cells resulted in *KRAS* knockdown with higher potency as compared to carrier-free AZD4785 (Fig. 4e).

Our data shows that AZD4785 recycling in EV can influence PS-ASO functional activity either by enhancing PS-ASO clearance from cells or by increasing intracellular delivery via EV-loaded AZD4785. To test the overall EV pathway contribution to PS-ASO functional activity, we used a small molecule inhibitor U18666A which blocks EV secretion by inducing cholesterol accumulation in LE[37,38]. In agreement with previous data, U18666A treatment induced the formation of an enlarged LE compartment with increased CD63 content in both cell lines (Fig. 5a and Supplementary Fig. S11a). Surprisingly, U18666A induced secretion of CD63+/CD81+EVs as detected by CD63 capture bead assay (Supplementary Fig. S12a, b). Next, we tested the effect of U18666A on AZD4785 functional activity (Table 2 and Fig. 5a, b). Again, we observed high AZD4785 potency in PC9 cells ($IC_{50} = 0.15$ μM) and low AZD4785 potency ($IC_{50} ≥ 28.4$ μM) in LK2 cells in the control conditions (Fig. 5b, c). Notably, U18666A improved AZD4785 efficacy ≈147-fold in LK2 cells ($IC_{50} ≥ 0.2$ μM) (Fig. 5b, c). A recent study indicated that U18666A also induces accumulation of LBPA[39] and we found that U18666A treatment increased intracellular LBPA content and number of LBPA+LEs in PC9 and LK2 cells (Supplementary Fig. S11c–f). To test the contribution of LBPA-dependent pathway, we used thioperamide maleate, an inverse agonist of the histamine H3 receptor HRH3 specifically increasing intracellular LBPA without trafficking pathways interfering[39]. We found that this novel compound induced accumulation of LBPA in PC9 and LK2 cells and increased number of LBPA+LE in LK2 cells only (Supplementary Fig. S11c–f). It also inhibited secretion of CD63+/CD81+EVs by PC9 cells and stimulated EV secretion by LK2 cells (Supplementary Fig. S12c, d). Interestingly,

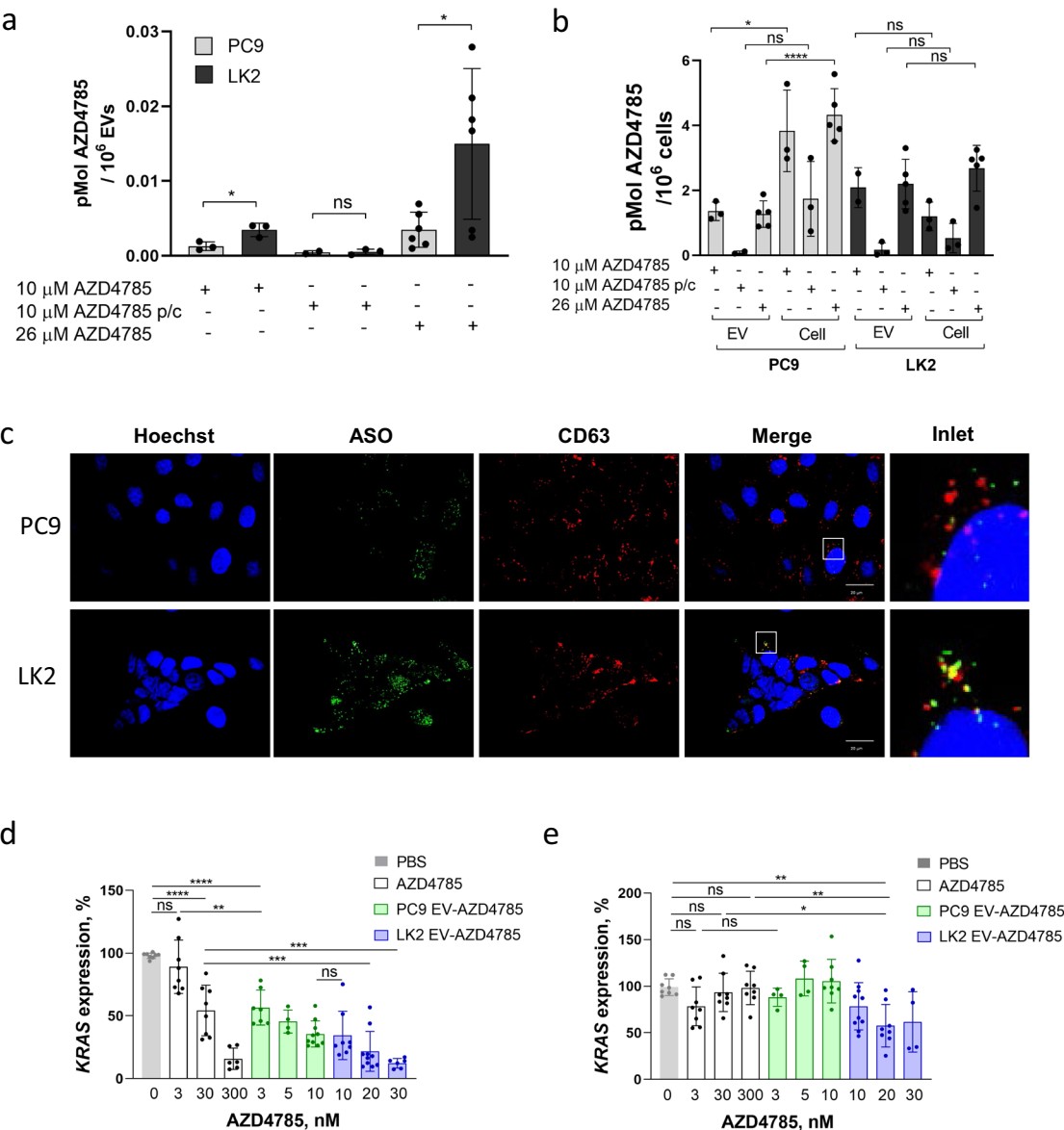

**Fig. 4 AZD4785 is recycled in EV and transferred between cells. a** Detection of AZD4785 in EVs. PC9 and LK cells were treated with AZD4785 for 24 h and EVs were isolated by differential ultracentrifugation. Quantification of AZD4785 in isolated EVs was conducted by UPLC-MS. t-test. *$p < 0.05$. $N = 3$–6. **b** Detection of intracellular AZD4785. AZD4785 was quantified in EV and cell lysates as in **a**. Intracellular and EV AZD4785 content were normalised per $10^6$ cells. **c** AZD4785 loaded in EVs is delivered to the cells. EVs were isolated from the conditioned media of the cells treated with 10 μM AZD4785 for 24 h as in **a**. PC9 and LK2 cells were incubated with AZD4785-loaded EVs for 3 h. Cells were washed, fixed and stained for CD63, AZD4785 and nucleus. **d**, **e** AZD4785-loaded EVs induce KRAS knockdown. AZD4785-loaded EVs were isolated from PC9 and LK2 cells after AZD4785 treatment as in **a** and aliquots with variable volume were added to PC9 (**d**) and LK2 (**e**) cells for 72 h. Error bars, standard deviation. ANOVA ****$p < 0.0001$. ***$p < 0.001$. **$p < 0.01$. ns non-significant. Data are duplicates, $N = 5$.

**Table 1 AZD4785 detection in plasma-derived EVs.**

| Treatment regime | Plasma collection time after the last dose | | | | | |
|---|---|---|---|---|---|---|
| | **72 h** | | | **168 h** | | |
| | **Plasma nM** | **EVs (fMol/$10^6$)** | **EVs plasma, nM** | **Plasma nM** | **EVs (fMol/$10^6$)** | **EVs plasma, nM** |
| PBS | Not detected | NA | NA | Not detected | NA | NA |
| AZD4785 125 mpk/wk | 12.48 | 0.07 | 0.85 | 4.28 | 0.03 | 0.11 |
| AZD4785 250 mpk/wk | 18.72 | 0.07 | 0.80 | 6.57 | 0.02 | 0.10 |

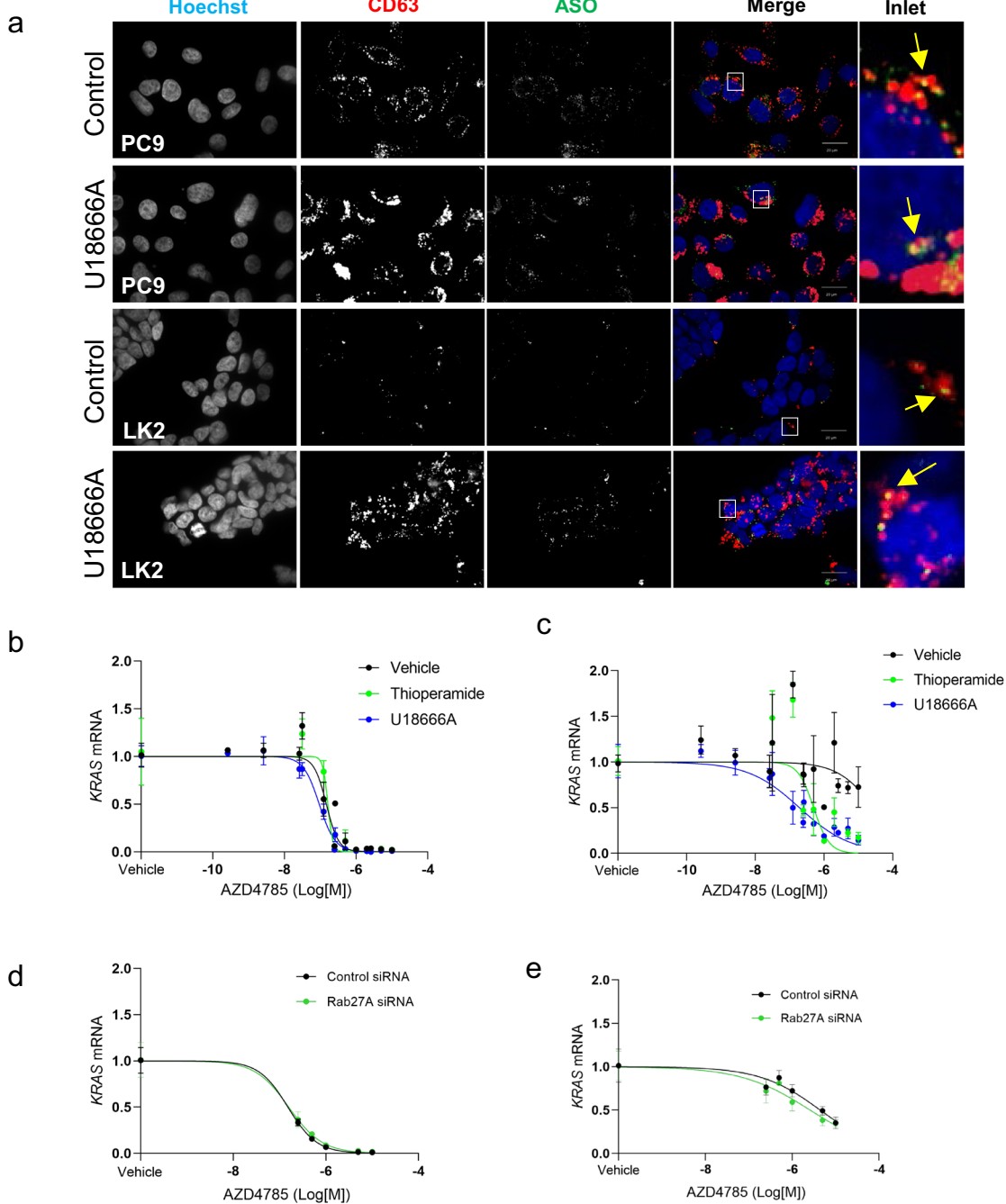

**Fig. 5 Inhibition of EV secretion and LBPA enhancement improve AZD4785 efficacy. a** Intracellular distribution of AZD4785 and CD63 after U18666A inhibitor treatment. PC9 and LK2 were treated with AZD4785 (26.1 μM) for 48 h, fixed and stained. LE and AZD4785 were stained with anti-CD63 and anti-PS-ASO antibodies, respectively, and visualised using secondary fluorescently labelled antibodies. Nuclei were stained with Hoechst 33342. To block EV secretion cells were treated with 7.1 μM U18666A 24 h prior to the addition of AZD4785. Cell images were acquired using Opera microscope (×60 objective, NA1.4). ($N = 3$). **b, c** KRAS expression in the inhibitor or siRNA-treated cells. PC9 (**b**) and LK2 (**c**) cells were treated with AZD4785 for 48 h. Cells were pre-treated with 7.1 μM U18666A or 10 μM thioperamide maleate (**b, c**) or control siRNA or Rab27a siRNA for 24 h (**d, e**) prior to AZD4785 treatment. KRAS expression was quantified by qPCR. Nonlinear regression analysis, each data point are duplicates from $N = 3$–9.

thioperamide maleate treatment improved AZD4785 productive uptake ≈56 fold times in LK2 cells ($IC^{50} \geq 0.5$ μM) but had no effect on PC9 cells (Fig. 5b, c and Table 2).

Intraluminal vesicles formed in LE compartment can also be secreted as exosomes[19] and to investigate the role of exosome-specific clearance, we knockdown Rab27a, a well-established regulator of exosome secretion in tumour cells[40] and used a small inhibitor of sphingomyelin phosphodiesterase 3 pathway, 3-O-Methyl-sphingomyelin[41]. Rab27A knockdown by using siRNA

resulted in the reduction of Rab27A to ≈15% level in both cell lines (Supplementary Fig. S12e). Importantly, Rab27A knockdown and SMPD3 inhibition reduced the secretion of CD63+/CD81+EV by LK2 cells and improved AZD4785 productive uptake ≈2 and ≈5.5 fold, correspondingly (Table 2, Fig. 5e and Supplementary Fig. S12h). Rab27a knockdown in PC9 cells had no effect on CD63+/CD81+ secretion as well as AZD4785 efficacy (Table 2, Fig. 5d and Supplementary Fig. S12f). Moreover, inhibition of sphingomyelin phosphodiesterase 3 pathway in PC9

**Table 2 IC$^{50}$ values for *KRAS* mRNA knockdown by AZD4785.**

| Treatment | IC$^{50}$, M |
|---|---|
| PC9 | |
| Vehicle | 1.473*10$^{-7}$ |
| Thioperamide | 1.631*10$^{-7}$ |
| U18666A | 0.935*10$^{-7}$ |
| Control siRNA | 1.516*10$^{-7}$ |
| Rab27a siRNA | 1.580*10$^{-7}$ |
| | IC$^{50}$ |
| LK2 | |
| Vehicle | 2.836*10$^{-5}$ |
| Thioperamide | 5.001*10$^{-7}$ |
| U18666A | 1.933*10$^{-7}$ |
| Control siRNA | 4.417*10$^{-6}$ |
| Rab27a siRNA | 2.581*10$^{-6}$ |

cells reduced the secretion of CD63+/CD81+EVs but had no effect on AZD4785 efficacy (Supplementary Fig. S12c, g).

## Discussion

ASO therapeutics in the clinic are currently delivering high ASO doses presumably to saturate bulk ASO non-productive uptake[7,8,42] or by adding cell-targeting ligands[43]. Both strategies rely on enhancing productive cell uptake as numerous studies have reported that the majority of PS-ASO is taken up into cells via the bulk, non-productive (lysosomal) pathway with only few drug molecules escaping LE compartment and reaching target RNA in the cytosol or nucleous[7,9,13–16]. Here we further investigated intracellular PS-ASO trafficking in good and poor productive uptake tumour cells and found that the productive uptake depends on level of PS-ASO delivery to the LBPA+LE compartment in both cell lines. Moreover, up to 50% of endocytosed AZD4785 which was delivered to CD63+LE compartment can be recycled via EV especially by LK2 which is a poor productive uptake line. Interestingly, we also show that the ASO recycling mechanism contributes to ASO intracellular clearance as well as intercellular ASO transfer in both cell lines. Importantly, modulation of LBPA-dependent and EV recycling pathways using small molecules enhances PS-ASO functional activity thus offering alternative strategies for improving ASO therapeutic applications.

Late endosomes (LE) are the central cargo sorting hubs merging endocytosis pathways and directing cargo for recycling to the plasma membrane, loading into the small intraluminal vesicles or delivering to lysosomes for degradation[18,19]. Lysosome's contribution to the non-productive ASO uptake and degradation were shown in multiple studies[9,10,13,17] and here we studied the role of LE in the ASO productive uptake. We selected two LE markers CD63 and LBPA[28,29,44] and these markers showed overlap in PC9 cells but were presented in the different LE populations in LK2 cells. Importantly, we found that in both cell lines the bulk AZD4785 uptake correlates with intracellular CD63 levels and endocytosed AZD4785 is detected in CD63+LE. On the contrary, LBPA staining revealed moderate colocalization in PC9 cells and no colocalization between LBPA and AZD4785 in LK2 cells. This striking difference in AZD4785 intracellular trafficking in PC9 and cells LK2 indicates that endocytosed PS-ASO somehow by-passing LBPA+LE sub-population in LK2 cells. LBPA is a key LE component assisting ASO endosomal escape[15] most likely by triggering intraluminal vesicles back-fusion with the LE membrane mechanism originally described for pathogens[28,45,46]. LBPA content in LE is high and reaches up to 15 mol% of all LE phospholipids in BHK cell line[28] yet the spatial

LBPA distribution across cellular LE population is still unknown[18]. LBPA LE level is regulated by Alix binding to the external LE surface[24] and Alix knockdown reduced LBPA LE content and ASO functional activity in A431 carcinoma cells[15]. We noted that Alix expression in LK2 cell was lower as compared to PC9 cells so it is tempting to speculate that Alix acts as an upstream regulator for LBPA spatial distribution across LE population in LK2 cells. Importantly, thioperamide maleate, an inverse agonist of the histamine H3 receptor HRH3 elevating intracellular LBPA content[39] and U18666A increased LBPA content in LK2 cells that resulted in significantly improved ($\approx$56 and $\approx$147-fold times, respectively) productive AZD4785 uptake. Altogether these data suggest that ASO delivery to LBPA-positive LE is a novel key factor delineating good and poor productive uptake cells. Although this finding is yet to be confirmed in a diverse panel of cells/tissues, it is tempting to suggest that the future ASO therapeutics can be particularly effective in the cells with high LBPA level, such as macrophages[47]. Alternatively, ASO can be applied in combination with the HRH3 agonists such as thioperamide maleate to boost endosomal escape and therapeutic effects.

What is the intracellular fate of AZD4785 delivered to CD63+LE? PS-ASO can be sorted for lysosomal degradation[9,10,13,17] but it remained unknown whether it can also be recycled in EVs and whether this also influences its functional activity. Endogenous RNA species can be secreted in EVs, which are generated in CD63+LE and secreted upon the fusion of LE with the plasma membrane[19,23,25,26] and here we report for the first time that the exogenous, formulation-free PS-ASO is recycled via EVs. Interestingly, secretion of endocytosed PS-ASO via unknown mechanism has been reported before[11]. Here we found that non-treated tumour cell lines PC9 and LK2 secret EVs with the similar size and proteomic composition and AZD4785 treatment induces EV secretion. Importantly, only EVs derived from AZD4785-treated LK2 cells were enriched with RNA-binding proteins and this unique profile can possibly be linked to the appearance of a large EVs population in response to ASO treatment. Although we detected the presence of endosomal markers, CD63 and Alix[31], the exact origin of these EVs remain to be discovered. To detect intact PS-ASO in EV, we deployed a novel mass spectrometry technique with a 10-fold improved lower detection limits (1 nM/ 0.005 $\mu$g/ml) as compared to previously reported assays such as capillary gel electrophoresis or ultrasensitive hybridisation enzyme-linked immunosorbent assay (ELISA) with detection limits of 0.06 $\mu$g/ml and 0.07 $\mu$g/ml ASO in plasma, correspondingly[8,48]. Interestingly, an improved sensitivity ultrasensitive hybridisation-based ELISA method has been recently reported for the phosphorodiamidate morpholino oligonucleotide[49] however oligonucleotides

length over 20–25 nucleotides is critical for sensitivity with this method[50]. Quantification of intact AZD4785 in the cells and in isolated EV using mass spectrometry revealed that LK2 secrete ≈3 times more AZD4785 per single vesicle as compared to PC9 (2103 and 9012 AZD4785 molecules per EV, correspondingly). Moreover, our data indicate that up to 50% of ASO can be recycled by LK2 cells as compared to intracellular levels, and this range is close to a recent study by Sahay et al. who established that up to 70% of siRNA delivered in lipid nanoparticles formulation can be recycled from the LE/MVB, presumably via exosomes[26]. We found that AZD4785 is recycled on the external EV surface, suggesting that PS-ASO can be recycled directly from the LE lumen or even bind to the secreted EVs. Yet, we observed dose-dependent EV AZD4785 loading and it correlated well with the intracellular levels, arguing against the hypothesis that PS-ASO binds to the EVs extracellularly. We identified 15 RNA-binding proteins in LK2 EVs and further tracking of PS-ASO and these proteins can potentially clarify EV loading mechanisms. Pulse-chaise experiment and detection of intact AZD4785 in plasma-derived EVs confirmed that the intact, full-size AZD4785 is actively recycled in EVs. AZD4785 wash-off ceased the secretion of the larger EVs population by LK2 cells further strengthening the link between AZD4785 endocytosis and EV secretion. To find out the exact contribution of the EV recycling pathway to ASO functional activity, we attempted to inhibit EV secretion pathway by blocking endosome maturation with U18666A[37,38] as well as by modulating Rab27A-dependent[40] and sphingomyelin phosphodiesterase 3-dependent[41] exosome biogenesis pathways. Interestingly, in agreement with previous reports U18666A treatment induced CD63 accumulation. Unexpectedly, it stimulated secretion of CD63+/CD81+EVs by both cell lines, most likely by inducing LBPA accumulation in LE and activation of recently-established novel LBPA/Alix EV secretion pathway[51]. Interestingly, in PC9 cells secretion of CD63+/CD81+EVs was Rab27A independent and inhibition of EV release with sphingomyelin phosphodiesterase 3 inhibitor did not influence AZD4785 potency. However, in LK2 cells Rab27A knockdown and inhibition of sphingomyelin phosphodiesterase 3 reduced CD63+/CD81+EV secretion and improved AZD4785 potency but only by ≈2 and ≈5.5 fold, correspondingly. Altogether these data strongly indicate mainly LBPA engagement in LE and to a less extent ASO clearance via the EV recycling pathway influence the productive ASO uptake in LK2 cells (Supplementary Fig. S13). Importantly, rapid AZD4785 delivery to LBPA+LE in PC9 cells and LBPA engagement dominates over EV recycling clearance pathway enabling good productive uptake hence future studies should be focused on the further understanding of the LBPA spatial distribution as well as the LBPA-mediated 'back-fusion' mechanism of endosomal escape (Supplementary Fig. S13). Modulation of intracellular PS-ASO trafficking by using small molecules regulating LBPA[15,39] or targeting Rab27A[40], SMPD3[41] or novel Alix-dependent[52] EV biogenesis pathways[40,41] are novel strategies overcoming non-productive uptake in select tumour cells and further analysis of EV role in ASO productive uptake across a varied panel of models and/or diseases are required in the future.

## Methods

**Cell culture**. PC9 (formerly known as PC-14) cells were obtained from ECACC General Cell Collection: (https://www.phe culturecollections.org.uk/products/celllines/generalcell/detail.jsp?refId=90071810&collection=ecacc_gc). LK2 cell was obtained from the Japanese Collection of Research Bioresources Cell Bank (https://cellbank.nibiohn.go.jp/~cellbank/en/search_res_det.cgi?ID=540). Both cell lines were authenticated by short tandem repeat (STR) analysis. Both cell types were grown at 37 °C and 5% of $CO_2$. The cells were grown in RPMI 1640 medium, supplemented with GlutaMAX™ (Gibco®) and 10% fetal bovine serum (FBS). Cell viability was measured using Trypan Blue solution or Vi-CELL™ XR Cell Viability Analyzer (Beckman Coulter, software v2.04). For EV isolation, cells were cultured in exosome-free medium prepared with RPMI 1640 medium, supplemented with GlutaMAX™, 10% exosome-depleted FBS (Fetal Bovine Serum, exosome-depleted, One Shot™ format (ThermoFisher Scientific, A2720803)) in the presence or absence

of AZD4785. For the pulse-chase experiment cells were treated with AZD4785 as above for 6 h, then washed 3 times with PBS and cultured in RPMI 1640 medium, supplemented with GlutaMAX™, 10% exosome-depleted FBS in the absence of AZD4785 for 24 h.

**Antibody, fluorescently labelled proteins, oligonucleotides and inhibitors**. Anti-Bovine Serum Albumin antibody [EPR12774] (Abcam, ab192603, 1:1000), Mouse Monoclonal antibody [clone 2C1] (IgG2b) to Human KRAS (LSBio Life-Span Bioscience Inc, LSB LS-C175665, 1:1000), Anti-Vinculin antibody (Abcam, ab73412, 1:2000), anti-human CD63 antibody (BD Pharmingen 556019, immunofluorescence 1:500, western blotting 1:1000), Alix Antibody (Cell signalling Technology, MA1-83977, 1:1000), rabbit anti-ASO (Ionis 13545, 1:25,000), Secondary Donkey Anti-Rabbit IRDye@800 CW (Li-COR 926-32213, 1:5000), secondary donkey anti-mouse IRDye@680 LT (Li-COR 926-68022, 1:5000), Donkey anti-Rabbit IgG (H + L) Alexa Fluor 568 (A10042, 1:200), Donkey anti-Mouse IgG (H + L) Highly Cross-Adsorbed Secondary Antibody, Alexa Fluor 488 (A21202, 1:200) and Goat anti-Mouse IgG (H + L) Highly Cross-Adsorbed Secondary Antibody, Alexa Fluor 647 (A21236, 1:200) (ThermoFisher Scientific), Albumin from Bovine Serum (BSA), Alexa Fluor™ 594 conjugate (ThermoFisher Scientific), U-18666A (Sigma, U3633), Thioperamide maleate salt (Sigma, T123), 3-O-Methyl-sphingomyelin (BML-SL225-0001, Enzo Life Sciences). Short interfering RNA (siRNA) oligonucleotides were ON-TARGETplus (HorizonDiscovery) non-targeting control pool (D-001810-10-05) and human Rab27A (L-004667-00-0005). Human Generation 2.5 KRAS PS-ASO (AZD4785/IONIS 651987, sequence GCTATTAGGAGTCTTT, capital letters are DNA, underlined letters are cEt-modified bases) was synthesised as previously described[53].

**Isothermal titration calorimetry (ITC)**. To evaluate the binding between BSA and the AZD4785 sequence, an Auto ITC-200 (Malvern Panalytical) was used. The affinity was measured in a 0.2 μm filtered buffers—PBS (pH 7.4), 10 mM PIPES (pH 7.4) containing 140 mM NaCl and 2.5 mM $CaCl_2$ or 10 mM PIPES (pH 6.15) containing 140 mM NaCl and 2.5 mM $CaCl_2$. AZD4785 (870 μM final concentration) was titrated into a BSA solution (200 μM). The latter was directly prepared for dissolution in a mixture of the buffer:water 30:1 to match the AZD4785 conditions. The data were collected at 25 °C, using 1 μl sequential injections every 4 min. Background level was derived from the heat of dilution of AZD4785 in the buffer and subtracted from the experimental values. Data were analysed using the 'One set of sites' model function in the Origin 7 software, which defines that all binding sites are equal. Each ITC experiment was performed in duplicates and data reported are averaged. Since multiple BSA seemed to bind onto one single ASO strand with, possibly, different types of interactions affinities, the stoichiometry was fixed to a 0.06 AZD4785:BSA ratio for all data analyses. This value was chosen as it corresponds to 1 BSA molecule for nucleotide and it allows for a reasonably good fitting. Since it was not possible to obtain the actual binding stoichiometry the calculated affinity values must be considered as a whole system value.

**Extracellular vesicles isolation and characterisation**. EVs were isolated from the conditioned medium using differential centrifugation as previously described[31,54] with modifications. Briefly, the conditioned medium was collected from cells (V = 60 ml) and centrifuged at 4 °C, 2000×g, 5 min using a Sorvall™ Legend™ XTR Centrifuge (ThermoFisher Scientific), with TX-1000 high capacity rotor, to remove live and dead cells, cell debris, and pellet larger vesicles such apoptotic bodies. The supernatant containing EVs was transferred to Thermo Scientific™ Sorvall™ High-Speed PPCO centrifuge tubes and centrifuged at 4 °C, 10,000×g, 30 min using a Thermo Scientific™ Sorvall™ RC-6 Plus Superspeed centrifuge with a Sorvall™ SS-34 Fixed Angle Rotor to remove large and medium-size EVs. The supernatant was transferred to Beckman Coulter Open-Top Thickwall polycarbonate tubes and ultracentrifuged at 6 °C, 100,000×g, 1 h using an Optima™ L-80 XP ultracentrifuge (Beckman Coulter) fitted with a Ti 70 rotor. The pellets were washed with PBS and ultracentrifuged again under the conditions described, before resuspending in a small aliquot (200–400 μl) of sterile filtered PBS for further analysis with NTA and fluorescence spectroscopy, as well as cell treatment or lysed (see below). An aliquot of EVs was used to determine AZD4785 content by MS and the number of EVs by NTA so we can calculate the dose of AZD4785 loaded in EVs. To study EV uptake, EVs were labelled with 5(6)-carboxyfluorescein diacetate N-succinimidel ester (Sigma-Aldrich, #21888) as previously described[55].

**Nanoparticle tracking analysis (NTA)**. Purified EV pellets were suspended in PBS and analysed by NTA using Nanosight model NS300 (Malvern Panalytical Ltd, Almelo, The Netherlands) equipped with a blue (488 nm) laser and sCMOS camera. Five 60 s videos were recorded, and the Brownian motion of particles was analysed using NanoSight NTA software v3.2 (Malvern, Worcestershire, United Kingdom) according to the manufacturer's protocol, with the screen gain at 10, camera level 14, focus −11, and detection level 7 to particles with minimal background.

**Inhibition of EV recycling pathway with inhibitors or siRNA**. For inhibition studies, PC9 and LK2 were cultured for 24 h in 96-well plates for qPCR or in T25

for western blotting analysis. Working solution of inhibitor (U18666A) was prepared using complete media as described above, added to cells, and incubated for 24 h before treatment with AZD4785. PC9 and LK2 cells were transfected with siRNA by reverse transfection using Lipofectamine™ RNAiMAX Transfection Reagent (ThermoFisher Scientific, 13778100) according to the manufacturer's protocol. In brief, to transfect cells in 96-well plate siRNA (3 pmol) was diluted in 25 µl Opti-MEM I Medium without serum and mixed with 0.25 µl Lipofectamine RNAiMAX and incubated for 15 min at room temperature. Cells were diluted in complete growth media and plated to each well containing siRNA-Lipofectamine transfection complex (125 µl, 25,000 cells per well) and incubated for 24–72 h.

**Cell and EVs lysis**. PC9 or LK2 cells were washed with PBS twice and then incubated with Accutase for 5 min at 37 °C and re-suspended in the complete media (RPMI 1640 medium, supplemented with GlutaMAX™ (Gibco®) and 10% exosome-depleted FB). Cell number was quantified using Vi-CELL™ XR Cell Viability Analyzer and cells were washed with PBS twice and spun down for 5 min at 322×$g$. Cell pellets were kept at −80 °C. Cells were lysed in RIPA Lysis and Extraction Buffer (Pierce, #89900) supplemented with Protease/Phosphatase Inhibitor cocktail (Cell Signalling Technology, 5872 S, 1:100) at the ratio 5*10$^6$ cells per 300 µl of lysis buffer. Next, cell lysates were ultrasound (Ultrasound Bath Grant, Power, 4 times × 10 s) and centrifuged 21,100×$g$ for 15 min at +4 °C. Supernatants were collected and kept at −80 °C until further analysis. EVs were lysed in RIPA Lysis and Extraction Buffer (Pierce, #89900) supplemented with Protease/Phosphatase Inhibitor Cocktail (Cell Signalling Technology, 5872S, 1:100) and centrifuged 21,100×$g$ for 15 min at +4 °C. Supernatants were collected and kept at −80 °C until further analysis. Protein concentration was measured using BCA protein assay (Pierce, #23225).

**Western blotting**. Samples were separated using Novex TM 4–12% Tris-Glycine PLUS Midi Gel (LifeTechnologies Corporation, WXP 41220) and transferred to Nitrocellulose membrane using iBlot Gel Transfer Stack (ThermoFisher Scientific, IB301001). Next, the nitrocellulose membrane was blocked in Odyssey Blocking buffer (PBS) (Li-COR, 927-400000) for 1 h at room temperature and incubated with primary antibodies diluted in PBS containing 0.1% tween-20 at +4 °C overnight. Then the membrane was incubated with Donkey Anti-Rabbit Secondary IRDye@800 CW (Li-COR, 926-32213) and Donkey Anti-Mouse Secondary IRDye@680 LT (Li-COR, 926-68022) and visualised using Odyssey ® CLx Infrared Imaging System.

**Dot blot analysis**. An aliquot of isolated EVs was absorbed onto nitrocellulose membranes for 30 min at room temperature and air dried. Next, membranes were blocked with PBS-0.1% BSA in PBS and incubated with primary antibodies (anti-CD63 1:500, anti-ASO 1:20,000) in the absence or presence 0.2% (v/v) Tween-20 at room temperature for 2 h. Membranes were washed 3 times in PBS, and were incubated with Donkey Anti-Rabbit Secondary IRDye@800 CW (Li-COR, 926-32213) and Donkey Anti-Mouse Secondary IRDye@680 LT (Li-COR, 926-68022), washed with PBS and visualised using Odyssey ® CLx Infrared Imaging System.

**Immunofluorescence staining**. Cells were washed with PBS three times and fixed in 4% (w/v) paraformaldehyde for 15 min at 37 °C. After incubation, cells were permeabilised with PBS-0.2% TX-100 for 5 min at RT. Cells were treated with blocking buffer (3% BSA in PBS) for 1 h at RT and incubated with primary antibodies at 4 °C overnight. After washing with blocking buffer, cells were incubated with fluorescently labelled secondary antibodies (1:200 in blocking buffer) at RT for 1 h in the dark. After removing the secondary antibodies, cells were washed with blocking buffer and PBS and incubated with 2 µg/ml Hoechst 33342 and 2 µg/ml HCS CellMask Blue Stain (Thermo Fisher Scientific, H32720) in PBS for 15–30 min at RT, and cells were washed twice with PBS. The plates were imaged using an Opera (PerkinElmer) Confocal High-Content Analysis System microscope (water ×60 objective (NA1.4) or air objective ×40 (NA 0.6)) and the image analysis was conducted in intensities and colocalization between ASOs and different organelles were analysed using Columbus (v 2.8.3.1266).

**RNA preparation and RT-qPCR**. Quantitative reverse transcription PCR (RT- qPCR) mRNA arrays were used to quantify *KRAS* and *Rab27A* mRNA. Total RNA was isolated using Qiagen (Hilden, Germany, 216413) FastLane Cell Probe Kit or Mag-MAX™-96 Total RNA Isolation Kit (ThermoFisher Scientific, AM1830) according to the manufacturer's instructions. One-step RT-qPCR using TaqMan® primer probe sets were performed following the manufacturer's (Qiagen) protocols with slight modifications. Briefly, 3 µl sample RNA was mixed with 0.5 µl of each primer probe set containing forward and reverse primers and fluorescently labelled probe (*KRAS* Taq-Man FAM-MGB, Hs00364284_g1, 18S rRNA Endogenous Control, VIC™/MGB probe, 4319412E, or *RAB27A* FAM-MGB Hs00608302_m1, ThermoFisher Scientific), 0.1 µl QuantiTect Probe RT Mix, 0.9 µl RNase-free water, and 5.0 µl QuantiTect Probe RT-PCR Master Mix in a 10 µl reaction. Reverse transcription was performed at 50 °C for 30 min, PCR initial activation step at 95 °C for 15 min, and 45 cycles of PCR at 94 °C for 15 s for denaturation and at 60 °C for 60 s for combined annealing/extension, using the

Applied Biosystems™ QuantStudio™ 12K or QuantStudio™7 Flex Real-Time PCR Systems (Applied Biosystems, USA). Relative gene expression was calculated by the 2(-Delta Delta C(T)) method as previously described[56]. IC$^{50}$ were calculated by nonlinear regression curve fit in GraphPad Prism 8 (v. 8.0.1(244)) using log (inhibitor) versus response, variable slope (four parameters) equation. The maximal *KRAS* expression (Top) and the maximal knockdown (Bottom) responses were constrained at 1 and 0, respectively.

**EV quantification by CD63 capture bead assay**. EV quantification in the cell culture media was performed as previously described with modifications[40]. In brief, anti-human CD63 antibody (35 µg) was immobilised to 4 µm aldehyde-sulfate beads (1 × 10$^8$), Aldehyde/Sulfate Latex Beads, 4% w/v, 4 µm (ThermoFisher Scientific, A37304) and CD63 capture beads kept in 0.5 mL of PBS containing 0.1% glycine and 0.1% sodium azide at 4 °C. PC9 and LK2 cells (100,000 cells/24 well plate/0.5 ml media) were treated in with inhibitors or siRNA as above in RMPI media supplemented with GlutaMAX™, 10% exosome-depleted FBS for 24 h. Conditioned media was collected and centrifuged for 5 min at 2000×$g$ and 30 min at 10,000×$g$ to remove cell debris and large EVs. Supernatants were incubated with CD63 capture beads (1 µl/0.4 ml media) on a shaker overnight at +4 °C. Cells were washed with PBS, detached from the plate by using Accutase and stained with 0.4% Trypan Blue (Gibco, 15250-061). Total number of live cells was counted using Cellaca MX Automated Cell Counter (Nexcelom Bioscience). Beads were washed with PBS supplemented with 2% BSA twice and stained with anti-CD81-PE antibody (1:50 in PBS supplemented with 2% BSA) for 1 h at room temperature, washed PBS supplemented with 2% BSA and analysed by using flow cytometry (LSR Fortessa, Beckton Dickinson, San Jose, CA). Single bead's population was gated, and arbitrary units were calculated as mean fluorescence units × percentage of positive beads and normalised to the number of live cells.

**Mice PC9 xenograft model and EV isolation from plasma**. Female SCID mice (Envigo) were housed under specific pathogen-free conditions in individually ventilated cages (Techniplast) at Alderley Park, United Kingdom. All animal experiments were approved by AstraZeneca animal welfare ethical review board and were conducted in 8- to 12-weeks-old female mice in full accordance with the UK Home Office Animal (Scientific Procedures) Act 1986. Mice were inoculated subcutaneously with PC9 cells (5 × 10$^6$ cells/mouse mixed with a 1:1 ratio in Matrigel (BD Biosciences)). Tumour growth was monitored twice weekly via calliper measurement and tumour volume was calculated using the equation: 3.14 × length × width$^2$/6000. Growing tumours were randomised and recruited onto study when tumours reached an average of ~0.1 cm$^3$ (~11 days after implant). AZD4785 was formulated using PBS and dosed at a dose volume of 0.1 ml/10 g subcutaneously. Animals selected onto study were dosed with either AZD4785 at 250 or 125 mpk/wk with dosing schedule 5 days on2/2 days off and once-a week dose for the groups '72 h' and '168 h', correspondingly. Control animals were given PBS once weekly. Animals were sacrificed 72 or 168 h post last dose and terminal blood samples (minimum 100 µl plasma) taken into K2EDTA coated tubes (SARSTEDT CB300 K2E). The tube was placed on wet ice and the plasma prepared within 30 min of sampling by centrifugation (1700×$g$ for 10 min at 4 °C). Plasma was then transferred into 0.5 ml Nunc cryotubes and immediately frozen until analysis. Plasma was thaw in the water bath (+37 °C) and was centrifuged 1 h 30 min at 47,000 RPM (rotor TLA-55, 4 °C), using Optima MAX-XP ultracentrifuge (Beckman Coulter). The pellets were washed with PBS and ultracentrifuged again and re-suspended in a small aliquot (60 µL) of sterile filtered PBS for further analysis.

**AZD4785 UPLC-MS analysis**. The snap-frozen samples were thoroughly defrosted before use and the samples prepared by centrifuging at 10,000 rpm for 5 min and taking 5 µl into a Greiner shallow well plate containing 45 µl of PBS solution. The sample was agitated for 5 min prior to being analysed directly by LC–MS using the conditions detailed below.

Samples were analysed by UPLC-MS utilising a Waters Xevo TQ-XS (WBA0259) and an Acquity UPLC system from Waters consisting of Sample Manager (M16UFL953M), Acquity PDA (F17UPD457A), Column Oven (E17CMP703G) and Binary Solvent Manager (E17BUR621G). The chromatographic conditions are as follows:

- 1 µl Injection.
- Flow rate 0.5 ml/min.
- Column = Phenomenex bioZen™ 3.6 µm intact XB-C8 100 × 2.1 mm.
- Column temp = 45 °C.
- Solvent A = 3.75 mM TFA: 100 nM HFIP in aqueous.
- Solvent B = 3.75 mM TFA: 100 nM HFIP in MeOH (50:50).
- Gradient = 80% A to 5% A in 4.0 mins and 2 min hold.
- Total run time = 6 min.

The Waters TQ-XS was operated in negative ion Electrospray (ESI) mode with the optimised transitions for AZD4785 (utilising the M-7H ion from the charge envelope) as shown in Table 3. Supplementary Figs. S3 and S4 show parent and daughter ion spectra, representative chromatography of the sample, linear

**Table 3 MRM Transition of AZD4785.**

| Compound | Mode | Parent ion | Daughter ion | Dwell | Cone voltage | Collision energy |
|----------|------|------------|--------------|-------|--------------|------------------|
| AZD0485 | ESI —ve | 773 | 360.5 | 0.329 | 20 | 30 |

regression and S/N at 5 nM.

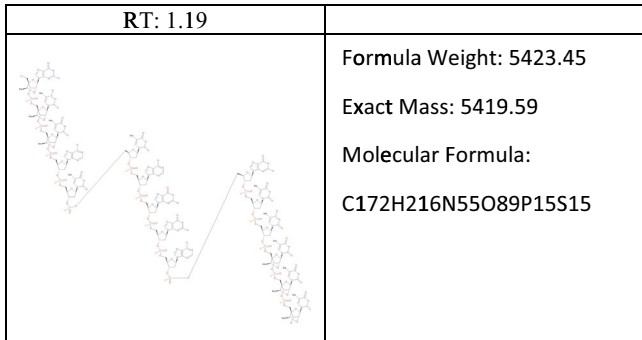

RT: 1.19

Formula Weight: 5423.45

Exact Mass: 5419.59

Molecular Formula:

C172H216N55O89P15S15

The chromatograms at each transition were extracted, smoothed and integrated to give the standard curve Chromatograms of the samples were treated similarly and by linear regression ($1/x$) an in-cell concentration was established in the re-suspended cell lysate using the Waters MassLynx TargetLynx™ product. The mean $r^2$ was observed to be 0.99 and the mean error of the QC samples of <10%.

To calculate the number of AZD4785 molecules per EV, the amount of AZD4895 per EV was multiplied by the Avogadro number (6.02214E + 23).

**Proteomics EV LC–MS/MS analysis.** Frozen pellets were re-suspended in 50 mM ammonium bicarbonate (Ambic, pH 8.0) to achieve a final concentration of approximately 2 μg/μl. Aliquots of 25 ul (~50 μg protein) of each sample were added to tubes containing 25 μl denaturation solution (8 M urea/50 mM AmBic, pH = 8). The resolubilised pellet was then reduced using 1 μl of 500 mM DTT and incubated at 56 °C for 45 min, followed by alkylation with 1 μl of 500 mM iodoacetamide; incubated for 45 min at 25 °C in the dark. All samples were diluted 4-fold with 50 mM Ambic to reduce the urea concentration (<2 M). Samples were digested overnight using Trypsin at a ratio of 1:50 (w/w) of enzyme to protein. Trypsin reaction was stopped by adding 1.3 μl TFA (0.5% to total volume) to each sample. Peptide purification was achieved with Waters Sep-Pak SPE cartridges. Cartridges were pre-conditioned and equilibrated with 100% Acetonitrile and 0.1% TFA, respectively. Trypsin-digested sample (50 μg) was loaded on to the cartridge and washed with 0.1% TFA (2 times) before eluting with 1 ml of 70% acetonitrile, 0.1% TFA solution. Eluted samples were then dried using SpeedVac and re-suspended in 100 μl 0.1%FA (~0.5 μg/μl). Peptides were separated and detected on a Waters nano-Acquity UPLC (M Class) coupled with Thermo Scientific Q-Exactive HF-X LC/MS system with nano-ESI source. 2 μl was injected and trapped on a Waters C18 Trap column (5 μm, 2 G 18 μm × 20 mm) for 3 mins at 5 μl/min, before being separated across a Waters peptide BEH C18 column, with a flow rate of 0.3 ml/min. Peptides were separated on a 120 min gradient, from 3 to 35% B (solvent A —0.1% formic acid in water; solvent B—0.1% TFA in acetonitrile). Peptides were acquired on a Thermo QExactive HF-X, operating in positive ion mode and data analysed in Proteome Discoverer v2.2.0.388. The mass spectrometry proteomics data have been deposited to the ProteomeXchange Consortium via the PRIDE partner repository with the dataset identifier PXD027804[57].

**Statistics and reproducibility.** For in vitro experiments at least three independent experiments were conducted with technical replicates. For experiments in vivo a minimum of 5 animals per treatment group was used. All experiments were measured in technical replicates, with a minimum of three biological replicates ($N = 3$). All attempts to replication of the described findings were successful. Statistical significance was analysed by one-way ANOVA for multiple group comparison or unpaired $t$-test for two groups comparison by using GraphPad PRISM (version 8.0.1(244)) software.

**Reporting summary.** Further information on research design is available in the Nature Research Reporting Summary linked to this article.

## Data availability

The authors declare that all relevant data supporting the findings of the study are available within the manuscript and its Supporting Information. Raw proteomic mass spectrometry data (Fig. 3E and Supplementary Figs. S5, S6, Supplementary Data 1, 2) have been deposited in https://www.ebi.ac.uk/pride/archive/ with the accession code PXD027804. Figure's raw data and unedited western blot images are included in the

Supplementary Data 3. All other data for all figures and results presented here are available from the corresponding author (ANK) upon reasonable request.

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

## Acknowledgements
We would like to thank Dr Giuditta Illuzzi for cell lysis protocol and useful suggestions, Dr Graeme Davies for the qPCR consulting, Dr Arpan S. Desai for editing the manuscript and technical advices, Dr Niek Dekker for the exosome consulting and Dr David Robinson for the assistance with cancer cell lines.

## Author contributions
A.K., S.R., E.C. and S.P. conceived the study, designed the experiments and wrote the paper, A.K. conducted cell culture experiments and purified EVs, P.D., D.L. and E.C. conducted mass spectrometry, analysed and interpreted the data, C.M. performed the microscopy and edited the paper, E.L. designed the experiments, interpreted data and edited the paper, L.H. performed the mice experiments, N.R., M.S., N.J.B. and P.W.D. conducted proteomics LC–MS/MS analysis, interpreted the data and edited the paper, S.S. performed Isothermal Titration Calorimetry and analysed the data, A.R. and A.R.M. provided ASO, discussed the data and edited the manuscript. All authors have read and approved the final manuscript.

## Competing interests
A.K., P.D., D.L., C.M., E.L., L.H., N.R., M.S., P.W.A.D., N.J.B., S.S., S.R., E.C. and S.P. are current or past employees and/or shareholders of AstraZeneca. A.R. and A.R.M. are employees and shareholders of Ionis Pharmaceutics.
