## [Transparent Peer Review File · Communications Biology]

Reviewers' comments:

Reviewer #1 (Remarks to the Author):

In this study A. Kapustin et al. investigated the fate of antisense nucleotide (AZD4785) in two cell that show either high or low AZD4785 uptake efficiency. They first demonstrated that internalized AZD4785 is loaded and recycled via extracellular vesicles, with an efficiency that is inversely proportional to the uptake efficiency. Drug-induced perturbation of EV-mediated recycling of AZD4785 increases the silencing of its target (KRAS). Then they show that EV-associated AZD475 is uptaken and delivered into acceptor cells where it silences the KRAS target. Interestingly, the delivery efficiency of EV-associated AZD4785 is proposed to be higher than the delivery efficiency of free-AZD475, although this last point is not highlighted in the summary. Finally, the authors speculate that manipulating (inhibiting) the secretion of "recycling" EVs may improve the efficacy of antisense oligonucleotides-based therapy.

Major comment:

This study has been initially rejected due to lack of originality, because oligonucleotides loading into EVs which mediate further horizontal transfer has been already proposed. However this study also shows important results that need to be fully considered and that seems like an invitation for further characterization.

To me, the most important result is that delivery and efficacy of AZD4785 is improved when ASOs are associated to EVs (figure 5). This raises the following question that the authors need to address: why then the so called "recycling/clearance" EVs that are proposed to contain ASO are not recaptured rapidly by the cell from which they emanate, in an autocrine fashion. Is it because EV (in particular EV-associated ASO) release is more efficient than EV capture and end delivery? This can surely be quantifiable through adequate pulse-chase experiments. Even in that case freshly secreted EV-loaded ASO should be uptaken more efficiently than free-ASO and should lead to a net gain within the cell. This is apparently not the case? Why? the authors should at least discuss in depth this apparent discrepancy, which will still need further experimental work, since this possibility is contradictory with the claim of the paper (exosome eliminate ASO from the cell).

In addition it is unclear if ASO is exclusively inside EV or can be at least partially bound to the EV-surface (see below).

Other Comments:

.Line 20 (introduction); the authors mentioned COPII recruitments on endosomes, as if it was a well-established mechanism. They may consider highlighting the fact that those observations/studies are very controversial.

.supplementary Fig1 and table 1; the authors investigated the effect on pH on ASO/albumin binding but limit the study to pH 7,4 or 6. Mature endosomes and lysosomes can be more acidic it would be interesting to test lower range of pH (down to 5 for instance).

In addition, it would be important to test ASO/EV-surface binding at the same pH. One possibility is that ASO bind the surface of EV (or acceptor cell) in a pH dependent manner. This would reinforce the study to characterize if ASO can be found at the surface of EV, and limit or increase the delivery to acceptor /target cells

. It is not clear if the majority of ASO is inside the EVs. The authors should carefully determine the ratio of intravesicular ASO /extravesicular ASO to demonstrate if the vast majority of ASO is indeed inside the vesicle (cytosolic side).

. Line 23 (discussion p2); the references used to mention back-fusion mechanism are not accurate. Authors should cite the initial work of J Gruenberg. In addition, such a fusion has been recently tested in vitro (Bonsergent et al 2019), this should be cited.

To summarize, the study provides interesting observations but seems too preliminary to fully support the claimed mechanism of action, unless the authors address (and rule out) the point mentioned above.

Reviewer #2 (Remarks to the Author):

Major/General Comments:

The idea that tumor cells use exosomes to resist ASOs and possibly other therapies is very intriguing and worthwhile studying. It is certainly a concept that would add another layer of complexity to how exosomes regulate tumor progression. While the authors demonstrate that the two cell lines studied have different sensitivities to the ASO, it is not completely clear from the data provided that this sensitivity is mediated by exosomes. There are also some inconsistencies in some of the data and characterization of MVBs and exosomes requires improvement. Critically, the experiments with combined AZD4785 and U18666A suggest that treating cells so that they retain the ASO and it can function to inhibit its target will not work and indicate that ASO recycling does not occur via an exosome biogenesis pathway, which is a main conclusion of the manuscript.

Overall, characterization of exosomes needs to be improved. EM is necessary to visualize exosomes along with NTA and more markers by immunoblot are required. Please see work from Clotilde Thery's lab published in PNAS 2016 for specifics on this. Further characterization of MVBs is also necessary, again by immunostaining for additional markers of LE/MVBs (not just CD63) and using EM to confirm in cells that one is observing an actual MVB. Functionally, knockdown of proteins involved in exosome secretion would ultimately verify MVBs/exosomes are being studied here – inhibiting Rab27 blocks fusion of MVBs with the plasma membrane so would produce an accumulation of MVBs (and possibly the ASO?) intracellularly, while decreasing exosomes and ASO carried in EVs if they are in fact carried by exosomes. The authors use U18666A as a way to functionally inhibit exosome release, but this drug is not commonly used in the EV field and the data provided with this drug are insufficient. A more conventional approach such as Rab27 knockdown should be used. It is important to realize that there are many overlaps between the pathways of exosome and microvesicle biogenesis, so they need to be properly characterized to conclude one over the other. If they cannot be well-characterized, it is necessary to stick to the more general extracellular vesicle term and acknowledge that either pathway may be involved. Please see review from Clotilde Thery in Nature Cell Biology 2019 for more information.

Despite in vivo work, which has been included in this study, those mouse experiments do not really seem to provide much support for their in vitro studies due to the inability to detect tumor exosomes in the xenografted mice and a lack of data correlating efficacy of the ASO treatment in vivo with the overall increase in exosome production in vivo. It seemed to be suggested that systemically, most or many cells will recycle the ASO in the mice? This set of experiments seems to end abruptly saying that tumor exosomes could not be detected in the plasma of the mice, but there are no further efforts to do so.

Specific comments:

- Fig 1, 2 A and B – quantify the colocalization of the ASO and LE/MVB. As they state in the results, it is visible in the images they show, but they should quantify this somehow, e.g., percentage of ASO signal that colocalizes with CD63 on a per cell basis, quantifying an appropriate number of cells. The methods actually states that colocalization was performed, but the data does not seem to be shown. An additional marker of LE/MVB is necessary too, as endosomes/MVBs are heterogeneous and additional markers might reveal more. Finally, TEM is necessary to confirm MVBs since the intraluminal vesicles of MVBs can only be visualized by TEM due to their small size.
- Fig 1, 2 B – it is not clear how they are looking at the various endocytosis pathways here as stated in the text. This sentence could be rephrased or just delete the words “via various endocytosis pathways” because it gives the impression that the authors will use different methods to inhibit specific endocytosis pathways in the following experiments, but they do not. The authors' reference in the text to the ASO occupying endosomes requires more data and additional stains for compartments of the endocytic pathway. Overall, characterization of endocytosis should be improved if the authors wish to describe the ASO as trafficking via endocytosis/endosomes. They cite previous work showing this, but it would be nice to see supplemental data here that ASO uptake in these particular studies is

proceeding as previously described.

- Fig 1, 2 B and C – the difference in these quantifications in panels B vs. C is not clear, mainly because it seems that all the signal is punctate, suggesting it is organized as loci. Can the authors clarify the difference here, highlighting in the images the different types of signals? Also, for both cell lines, there appears to be a noticeable increase in ASO loci with increasing dose of the ASO based on the images, but that does not come up with the quantifications in the graphs in panels B or agree with the statement in the text regarding saturation. Again, a clarification of the quantification would help to explain this along with a single color high power image of the ASO staining and arrows indicating loci. The importance of overall CD63 intensity shown in panels C is not clear, as presumably it is the perinuclear, punctate CD63 that is associated with LE/MVB, yet no change in loci in panels B is observed for either cell line. Moreover, it is not apparent from the images that the overall CD63 intensity actually increases as shown in the graph. Can this data in C be verified by immunoblot for both cell lines?
- Fig 3A and B – TEM images of the exosomes are necessary along with NTA measurements. The authors should be careful not to describe the EV population they are studying as exosomes based solely on NTA size measurement since microvesicles that bud from the plasma membrane can also be of this small size.
- Fig 3c – the authors comment in the text that PC9 cells secrete 2x more exosomes than LK2 cells, but they do not provide a statistical comparison of this in the graph in 3c.
- Fig 3e – the authors need to show additional markers by western. Previous work from Clotilde They published in PNAS has suggested that it is necessary to confirm the presence of CD9 and CD81 along with CD63 to state that the EVs being studied are bona fide exosomes. Other helpful markers cited in work from her lab are TSG101 and syntenin.
- Fig 3F and G – it is really intriguing that the authors found the LK2 cells packaged more of the ASO into exosomes than the PC9 cells, which fits with the increased potency of the ASO in the PC9 cells. However, it is not clear that this increased packaging is occurring based on the data in in Figs 1 and 2. It would be expected that the LK2 cells have increased localization of the ASO to LE/MVB, but that is not clear from the data provided as a colocalization was not performed, and the LK2 cells do not have increased ASO loci, which also might be expected if it is localized and packaged into MVBs at a greater rate, compared to PC9 cells.
- Fig S2B – in the text the authors set out to verify if the EVs they have collected from mouse plasma are derived from the human PC9 xenograft and they conclude from the data that the EVs are not from the xenograft. However, they do not go on to further address by additional methods whether they are able to detect EVs from the xenograft tumor in the mice. Do the authors think this means that overall, systemic administration of the ASO affects EV production in cells throughout the mice as they describe for the data shown in 3H and 3I? This would still agree with some of their in vitro work suggesting that in general, the ASO can affect exosome production, but it complicates the ability to confirm this is happening in the tumor cells themselves in vivo, which is important for understanding how well the ASO can target Kras in the tumor cells. Can they also perform this study with the LK2 cells and use immunofluorescence of tumor sections to look at how much of the ASO is retained in the LK2 vs PC9 cells? Or measure the ASO in a piece of tumor, assuming there are not too many other cell types present and it is mostly tumor cell mass.
- Fig 4 – It is critical to show that this U18666A inhibits exosome secretion; this data does not appear to be present. Also, it seems surprising that the ASO does not appear to accumulate in the cells when U18666A is used. If the ASO is being packaged trafficked to LE/MVB and packaged for secretion as exosomes, it would be expected that this drug would promote its accumulation in the cells since they cannot recycle it through exosomes. Can the authors explain this? Also, this drug seems to differentially affect CD63 loci/intensity/loci size between the two cell lines even in basal conditions without the ASO. Why is this? This finding raises questions about how robust this drug is and suggests another approach should be used.
- Fig 5 C and D – this data questions the importance of this pathway in helping LK2 cells resist the ASO because use of U18666A barely rescues the ability of AZD4785 to decrease KRas expression in LK2 cells. Again, maybe another approach targeting exosome secretion if this pathway is important can provide a better result.

Reviewers' comments:

Reviewer #1 (Remarks to the Author):

In this study A. Kapustin et al. investigated the fate of antisense nucleotide (AZD4785) in two cell that show either high or low AZD4785 uptake efficiency. They first demonstrated that internalized AZD4785 is loaded and recycled via extracellular vesicles, with an efficiency that is inversely proportional to the uptake efficiency. Drug-induced perturbation of EV-mediated recycling of AZD4785 increases the silencing of its target (KRAS). Then they show that EV-associated AZD475 is uptaken and delivered into acceptor cells where it silences the KRAS target. Interestingly, the delivery efficiency of EV-associated AZD4785 is proposed to be higher than the delivery efficiency of free-AZD475, although this last point is not highlighted in the summary. Finally, the authors speculate that manipulating (inhibiting) the secretion of "recycling" EVs may improve the efficacy of antisense oligonucleotides-based therapy.

We thank the reviewer for nicely summarising the major findings. We investigated ASO trafficking and EV recycling further and included novel data supporting our previous conclusions and showing that the majority of ASO is rapidly delivered to CD63 late endosomes (Fig. 1) and recycled in EVs (Fig 3). However, ASO functional activity is limited by the distribution of LBPA which facilitates ASO escape from late endosomes (Fig. 2, Fig 5B-E). We also investigated EV proteomic composition and found that the enhanced ASO recycling in EV is associated with the increased secretion of RNA binding proteins (Figs. 3A, S5, S6 and Table S1). These findings are summarised altogether in the Fig. S12.

Major comment:

This study has been initially rejected due to lack of originality, because oligonucleotides loading into EVs which mediate further horizontal transfer has been already proposed. However this study also shows important results that need to be fully considered and that seems like an invitation for further characterization.

We appreciate this comment and agree that the oligonucleotide transfer in EVs was proposed earlier and demonstrated for multiple endogenous RNA species (miRNA etc) or RNA delivered in lipid nanoparticles. We have provided the relevant references that cite these important studies (refs 20, 24, 26, 27). However, we provide the first evidences of the recycling of exogenously added formulation free antisense oligonucleotides along with the role of LBPA and albumin. We have also shown the ability of exogenously manipulating these pathways with small molecules to redirect the payload and overall target modulation in vitro. Finally, we are providing advanced intracellular quantification using highly sensitive ASO UPLC-MS analysis which to our knowledge has not been attempted in this context previously. Using quantitative approach, we were able measure intracellular and EV ASO concentration showing the recycling efficacy.

To me, the most important result is that delivery and efficacy of AZD4785 is improved when ASOs are associated to EVs (figure 5).

We thank the reviewer for stressing the novelty of our results and quantitative approach which allowed to compare directly the functional efficacy of free and EV-associated ASO. We were interested to find out what are the key factors limiting ASO pharmacological activity in tumour cells which was found to be very different compared to hepatocytes and liver delivery. Following reviewer's comments, we summarise that the combination of two limiting steps affecting ASO endosomal escape and activity: (1) ASO delivery to the LBPA-positive late endosomal compartment and (2) ASO recycling in EVs (Fig. S12).

This raises the following question that the authors need to address: why then the so called "recycling/clearance" EVs that are proposed to contain ASO are not recaptured rapidly by the cell from which they emanate, in an autocrine fashion. Is it because EV (in particular EV-associated ASO) release is more efficient than EV capture and delivery? This can surely be quantifiable through adequate pulse-chase experiments. Even in that case freshly secreted EV-loaded ASO should be uptaken more efficiently than free-ASO and should lead to a net gain within the cell. This is apparently not the case? Why? the authors should at least discuss in depth this apparent discrepancy, which will still need further experimental work, since this possibility is contradictory with the claim of the paper (exosome eliminate ASO from the cell).

We thank the reviewer for this important question how ASO entrapped in EVs can contribute to intracellular ASO pool and ASO functional activity. We included additional data into revised

manuscript showing that ASO functional activity depends on the (1) ASO delivery to LBPA LEs and (2) EV recycling contributing together:

a) ASO bulk delivery in EVs to cells is more efficient as compared to free ASO intracellular ASO delivery as compared to free ASO (Fig. 4C, Fig. S7A, Fig. S7B).

b) The majority of free or EV-associated ASO is delivered to CD63 late endosomes in PC9 and LK2 cells (Figs 1 A-H, Fig. 4C). However, ASO functional activity is only achieved upon delivery to late endosomes containing LBPA and this was only observed in PC9 cells but not in LK2 cells (Figs 2 A-D).

c) CD63 and LBPA are presented in the same late endosome population in PC9 cells (where ASO has high activity) and are separated in two different populations in LK2 cells (ASO resistant) (Fig. S1C and S1D).

d) Modulation of EV recycling pathways using U18666A or Rab27a siRNA and LBPA level improved ASO activity in LK2 cells but not in PC9 cells (Figs 5B-E).

Hence, there are two key limiting factors – the presence of LBPA in late endosomal compartment and recycling in EV that regulates ASO activity in ASO-resistant cell line (LK2). This is highlighted by the new manuscript title (Antisense oligonucleotide activity in tumour cells is influenced by intracellular LBPA distribution and 5 extracellular vesicles recycling). We proposed the hypothesis that the combination of these factors define ASO endosomal escape and activity so inhibiting EV recycling on its own may not be sufficient without increasing LBPA levels (Fig. 12S).

In addition it is unclear if ASO is exclusively inside Ev or can be at least partially bound to the EV-surface.

We thank reviewer for this comment and additional experiments/data are highlighted below (see below and Fig. S3).

Other Comments:

.Line 20 (introduction); the authors mentioned COPII recruitments on endosomes, as if it was a well-established mechanism. The may consider highlighting the fact that those observations/studies are very controversial.

We agree that COPII hypothesis requires further investigation however our introduction highlights few established intracellular trafficking mediators that regulate ASO functional activity. We provided corresponding reference so the readers can refer to original study (Liang XH et al., Nucleic Acid Res 2018). Can we please ask the reviewer to suggest the reference opposing the hypothesis of COPII contribution to ASO endosomal escape?

Supplementary Fig1 and table 1; the authors investigated the effect on pH on ASO/albumin binding but limit the study to pH 7,4 or 6. Mature endosomes and lysosomes can be more acidic it would be interested to test lower range of pH (down to 5 for instance).

We agree with the reviewer that the lysosomal range of pH is the most relevant. However, due to the limitation of the analytical technique in the low pH range it is impossible to measure under acidic conditions. Our further investigation revealed that the buffer acidification induces albumin denaturation so isothermal titration calorimetry data were inaccurate in this range. Hence, we removed Table 1 and Fig. 1S and included ICT data obtained at neutral pH only (Fig. S3C). We found the low affinity interaction between ASO and albumin and these data strongly indicate that albumin cannot be involved in ASO tethering to EVs.

In addition, it would be important to test ASO/EV-surface binding at the same pH. One possibility is that ASO bind the surface of EV (or acceptor cell) in a pH dependent manner. This would reinforce the study to characterize if ASO can be found at the surface of EV, and limit or increase the delivery to acceptor /target cells

We thank the reviewer for this comment. We agree that ASO localisation in EV is very important for mechanistic insights. Our data showed that majority of ASO is bound to EV surface (see below). However, we did not investigate ASO-EV interaction further as it was out of the study

scope. This study may also be technically challenging due to the partial protein denaturation at the acidic pH values.

It is not clear if the majority of ASO is inside the EEVs. The authors should carefully determine the ratio of intravesicular ASO /extravesicular ASO to demonstrate if the vast majority of ASO is indeed inside the vesicle (cytosolic side).

We thank reviewer for this interesting insight. Surface ASO localisation would indicate that ASO recycling occurs along with the cell surface receptors. Intraluminal EV localisation would indicate loading of ASO from the cell cytosol (similar to the EV loading with miRNAs and other endogenous RNA species).

We defined ASO localisation by using dot blot and found that the majority of ASO associated with the EV surface (Fig. S3A) suggesting that most of exogenous ASO remains associated with the cell surface receptors. This is novel way of oligonucleotide recycling and it is different from the secretion of the endogenous nucleic acids. However, it can also be explained by non-specific ASO binding to EV outside of the cells. To exclude this mechanism, we conducted pulse-chase experiment and found that ASO can be detected in the secreted EV after ASO removal from the media (Figs. 4A and 4B). Notably, we observed the dose dependent EV loading with ASO which correlated well the intracellular ASO levels (Fig. 4A). Moreover, we identified the great impact of ASO treatment on EV size, concentration as well as proteomic composition (Figs 3A-C, Fig. 3E, Fig. S2, Fig. S5, Fig. S6, Tables S1 and S2). These data support the hypothesis that ASO at least partially is recycled with the membrane surface receptors and these receptors are yet to be identified in the future studies.

Line 23 (discussion p2); the references used to mention back-fusion mechanism are not accurate. Authors should cite the initial work of J Gruenberg. In addition, such a fusion has been recently tested in vitro (bonsergent et al 2019), this should be cited.

We appreciate this important comment and included original references for studies where the back-fusion mechanism was firstly proposed (Prof Jean Gruenberg laboratory studies, ref 29 and 45). EV content release assay invented by Emeline Bonsergent and Gregory Lavieu is an important advancement in the understanding of EV delivery mechanism to the target cells so we included this reference to the study (ref 46).

To summarize, the study provides interesting observations but seems too preliminary to fully support the claimed mechanism of action, unless the authors address (and rule out) the point mentioned above.

We appreciate insights and comments by the reviewer. We addressed the mechanisms of ASO resistance in tumour cells and found that the combination of two factors – ASO delivery to LBPA late endosomes and EV recycling play the key role. We also agree that the mechanisms of ASO EV transportation between cells should be investigated in the future studies.

Reviewer #2 (Remarks to the Author):

Major/General Comments:

The idea that tumor cells use exosomes to resist ASOs and possibly other therapies is very intriguing and worthwhile studying. It is certainly a concept that would add another layer of complexity to how exosomes regulate tumor progression. While the authors demonstrate that the two cell lines studied have different sensitivities to the ASO, it is not completely clear from the data provided that this sensitivity is mediated by exosomes. There are also some inconsistencies in some of the data and characterization of MVBs and exosomes requires improvement.

We thank the reviewer for this insightful comment. Future ASO application for treating cancer depends on identifying barriers and variability in tumour responses to ASO therapeutics, ranging from functional and productive to non-responsive (resistant). This mechanistic insight would be critical in screening cell models for predicting degree of response to ASO therapeutics.

Following the reviewer's comment, we included additional data showing that the combination of two factors, namely the delivery to LBPA late endosomes and extracellular vesicles (EVs) recycling contribute to ASO resistance across 2 tumour cell lines:

(1) we characterised further LE/MVBs ASO delivery using novel late endosomal marker, LBPA, which is critically important for ASO endosomal escape. We found that ASO delivery to CD63+/LBPA+ is associated with high ASO activity in PC9 cells. However, in the resistant cell line, LK2, ASO is delivered to CD63+ late endosomes lacking LBPA (Figs. 2 A-D, Fig. S1 C-D).

(2) We investigated proteomic composition of extracellular vesicles secreted in different conditions (+/- ASO) and found markers for various EV populations including exosomes and microvesicles (Figs 3E, S5, S6 and Table S1). Therefore, we replaced the term "exosomes" to more correct term "extracellular vesicles". Moreover, we identified 15 RNA-binding proteins which were exclusively secreted by LK2 cells treated with ASO (Fig. S6 and Table S2) and this correlates well with higher ASO content in LK2-derived EVs (Fig. 4A).

(3) We tested two additional inhibition approaches - Rab27a siRNA to block exosome secretion and thioperamide maleate to enhance intracellular LBPA level (Fig. S11). Our data indicate that these approaches improves ASO sensitivity in LK2 but not in PC9 cells thus linking EV secretion and ASO sensitivity (Figs. 5B-E).

We concluded that there are two key limiting factors regulate ASO activity in ASO-resistant cell line (LK2) – the presence of LBPA in late endosomal compartment and recycling in EV. This is reflected in the revised manuscript title ("Antisense oligonucleotide activity in tumour cells is influenced by intracellular LBPA distribution and extracellular vesicles recycling"). We proposed the hypothesis that the combination of these factors define ASO endosomal escape and activity so inhibiting EV recycling on its own may not be sufficient without increasing LBPA levels (Fig. 12S).

Critically, the experiments with combined AZD4785 and U18666A suggest that treating cells so that they retain the ASO and it can function to inhibit its target will not work and indicate that ASO recycling does not occur via an exosome biogenesis pathway, which is a main conclusion of the manuscript.

We thank the reviewer for this insightful question. Our novel data characterising ASO intracellular trafficking provides an explanation as to why U186666 inhibition is only improving ASO functional efficacy in LK2 cells (Figs. 5B and 5C) which was an important finding to establish differences in responses between the 2 cell lines and underlying cellular machinery and pathways. We found that ASO functional activity is defined by the trafficking through LBPA-positive late endosomes in both cells prior to EV secretion. We found that the bulk internalised ASO is trafficked through the CD63 LE in both cell lines, PC9 and LK2 (Fig. 1). However, LBPA colocalise with CD63 and ASO in PC9 cells only (Fig. 1 and Fig. S1). The majority of internalised ASO is recycled by LK2 cells in EVs (Fig. 4) and inhibition of EV release with U186666A and Rab27a using siRNA enhanced ASO activity in LK2 cells (Fig. 5C and 5E) most likely by affecting upstream LBPA-driven pathways (Fig. S11). Hence we hypothesized that both factors – trafficking through LBPA+ late endosomes and EV recycling influence ASO functional activity in the ASO resistant cells and this mechanism is summarised in Fig. S12.

Overall, characterization of exosomes needs to be improved. EM is necessary to visualize exosomes along with NTA and more markers by immunoblot are required. Please see work from Clotilde Thery's lab published in PNAS 2016 for specifics on this.

We thank the reviewer for this comment and agree that TEM would be a good additional visual characterisation of vesicles to use the term "exosomes". However, we conducted deep proteomic study of extracellular vesicles secreted in different conditions (+/- ASO) and found markers for various EV populations including exosomes and microvesicles (Figs 3E, S5, S6 and Tables S1 and S2). Given the Clotilde Thery's recommendations (PNAS, 2016, REF 32) and our EV characterisation approaches (NTA, Western blotting, proteomic study) we replaced the term "exosomes" to more general term "extracellular vesicles".

Further characterization of MVBs is also necessary, again by immunostaining for additional markers of LE/MVBs (not just CD63) and using EM to confirm in cells that one is observing an actual MVB.

We thank the reviewer for this insightful suggestion, and we conducted an additional study of intracellular ASO trafficking. Firstly, we investigated ASO delivery to CD63 late endosomes by using single cell analysis and found direct correlation between ASO uptake and CD63 level in both cell lines (Fig. 1G and 1H). These data indicate that the bulk ASO is delivered to CD63 late endosomes. To identify late endosomes population contributing to ASO endosomal escape and functional activity we selected novel marker LBPA which is also exclusively presented in late endosomes (Prof Jean Gruenberg laboratory studies, refs 19, 29, 30 and 45). Prominently, we found that CD63 and LBPA are presented in the same late endosome population in PC9 cells (where ASO has high activity) and are separated in two different populations in LK2 cells (ASO resistant cells) (Fig. S1C and S1D). Moreover, we observed ASO delivery to LBPA late endosomes in PC9 cells but not in LK2 cells (Fig. 2A and 2D). We agree with the reviewer that the EM is the definitive method to identify multivesicular bodies in the cells indisputably. However, we found that tumour cells release heterogenous EV population so the term "exosomes" was replaced with the more correct term "extracellular vesicles" (Figs 3E, S5, S6 and Tables S1 and S2) Hence, identifying exact EV populations may be conducted in the future studies based on the novel markers that we are reporting in our manuscript.

Functionally, knockdown of proteins involved in exosome secretion would ultimately verify MVBs/exosomes are being studied here – inhibiting Rab27 blocks fusion of MVBs with the plasma membrane so would produce an accumulation of MVBs (and possibly the ASO?) intracellularly, while decreasing exosomes and ASO carried in EVs if they are in fact carried by exosomes. The authors use U18666A as a way to functionally inhibit exosome release, but this drug is not commonly used in the EV field and the data provided with this drug are insufficient. A more conventional approach such as Rab27 knockdown should be used. It is important to realize that there are many overlaps between the pathways of exosome and microvesicle biogenesis, so they need to be properly characterized to conclude one over the other. If they cannot be well-characterized, it is necessary to stick to the more general extracellular vesicle term and acknowledge that either pathway may be involved. Please see review from Clotilde Thery in Nature Cell Biology 2019 for more information.

Clotilde Thery summarised the key EV secretion pathways, EV markers and EV uptake mechanisms in the review (Mathilde Mathieu et al., 2019). In fact we found most recent follow up study from Clotilde Thery's laboratory describing the novel markers for plasma membrane vs late endosome - derived EV population. This study is cited in our manuscript (ref 33). Given these recent data and the reviewer's comment we replaced term "exosomes" to the more general term "extracellular vesicles" based on our novel proteomic data (Figs 3E, S5, S6 and Table S1). Therefore, in our study we used an established approach to block LE fusion to the plasma membrane published elsewhere (ref 38 and 39) to inhibit bulk EV production by cells rather than blocking specific pathways. We validated this approach by observing the clustering of CD63 late endosomes (Fig 5A). Although exosomes represent heterogenous population (refs 32 and 33) we also confirmed the presence of Alix and CD63 (Fig. 3D) and tested Rab27a siRNA approach and showed that it improved ASO activity in LK2 cells (Fig. 5E). Our additional data showed that the delivery to LBPA late endosomes is an upstream bottle-neck stage and just blocking downstream EV secretion by PC9 cells did not bring any benefits as oppose to blocking EV release in the resistant LK2 cells (Figs 5B-E). The cross-talk between LBPA and CD63 late endosomes in the resistant cells and identifying of exact EV population involved in ASO recycling are the subject of further studies.

Despite in vivo work, which has been included in this study, those mouse experiments do not really seem to provide much support for their in vitro studies due to the inability to detect tumor exosomes in the xenografted mice and a lack of data correlating efficacy of the ASO treatment in vivo with the overall increase in exosome production in vivo. It seemed to be suggested that systemically, most or many cells will recycle the ASO in the mice? This set of experiments seems

to end abruptly saying that tumor exosomes could not be detected in the plasma of the mice, but there are no further efforts to do so.

We thank the reviewer for this important comment. We agree with the suggestion that other tissues can recycle ASO in EVs and contribute to ASO PK/PD profile. Here we isolated EV from mice plasma by using ultracentrifugation and characterised these EVs by NTA and western blot (Fig S10). Importantly, we established that ASO can be recycled in EV in vivo (Table 1). We also quantified ASO distribution between the free and EV-associated forms (Table 1). These data indicate that up to 5% of plasma ASO can be circulating in EV. We agree with the reviewer that we provided only limited EV characterisation and we could not observe tumour derived EVs in the isolated pellets (Fig. S10). Identifying the origin of these EVs and/or the role of these EV in tumour resistance would need to be done in the future. Given the improved efficacy of EV-associated ASO in vitro (Fig. 4D and 4E) we assume that even this little amount has high clinical significance and this needs to be investigated further.

Specific comments:

- Fig 1, 2 A and B – quantify the colocalization of the ASO and LE/MVB. As they state in the results, it is visible in the images they show, but they should quantify this somehow, e.g., percentage of ASO signal that colocalizes with CD63 on a per cell basis, quantifying an appropriate number of cells. The methods actually states that colocalization was performed, but the data does not seem to be shown. An additional marker of LE/MVB is necessary too, as endosomes/MVBs are heterogeneous and additional markers might reveal more. Finally, TEM is necessary to confirm MVBs since the intraluminal vesicles of MVBs can only be visualized by TEM due to their small size.

We thank the reviewer for this important note. We quantified Pearson's colocalization coefficient for CD63 and ASO colocalization (Fig. 1F) and analysed the correlation between ASO and CD63 spots by single cell analysis (Fig 1G and 1H). This data supported that bulk ASO uptake correlates with CD63. Next, we characterised colocalization with another LE marker, LBPA and found colocalization in PC9 but not in LK2 cells (Fig. 2A, Fig. 2D). We think it is a very significant advantage in the field that can be used to identify ASO resistant cells in the future. Based on our novel proteomic data (Figs 3E, S5, S6 and Table S1) we replaced the term "exosomes" to "extracellular vesicles" so identifying exact late endosomal population involved in ASO EV recycling is the next step.

- Fig 1, 2 B – it is not clear how they are looking at the various endocytosis pathways here as stated in the text. This sentence could be rephrased or just delete the words "via various endocytosis pathways" because it gives the impression that the authors will use different methods to inhibit specific endocytosis pathways in the following experiments, but they do not. The authors' reference in the text to the ASO occupying endosomes requires more data and additional stains for compartments of the endocytic pathway. Overall, characterization of endocytosis should be improved if the authors wish to describe the ASO as trafficking via endocytosis/endosomes. They cite previous work showing this, but it would be nice to see supplemental data here that ASO uptake in these particular studies is proceeding as previously described.

We thank the reviewer for this comment, and we deleted this sentence accordingly. As we stated above, we tracked ASO delivery to the late endosomes, an organelle where ASO endosomal escape occurs (ref 15). We exploited two markers, CD63 and LBPA and found that the bulk internalised ASO is delivered to CD63 late endosomes (Fig. 1G and 1H) but functional activity related to the delivery to LBPA late endosomes. (Fig. 2A and 2D and Fig. S1C and S1D). This mechanism is summarised in Fig. S12. ASO intracellular trafficking to early endosomes and lysosomes has been reported by our laboratory before (ref 18) and these data can be included in the supplement if required.

- Fig 1, 2 B and C – the difference in these quantifications in panels B vs. C is not clear, mainly because it seems that all the signal is punctate, suggesting it is organized as loci. Can the authors clarify the difference here, highlighting in the images the different types of signals? Also, for both cell lines, there appears to be a noticeable increase in ASO loci with increasing dose of the ASO based on the images, but that does not come up with the quantifications in the graphs in panels B or agree with the statement in the text regarding saturation. Again, a clarification of the quantification would help to explain this along with a single color high power image of the ASO

staining and arrows indicating loci. The importance of overall CD63 intensity shown in panels C is not clear, as presumably it is the perinuclear, punctate CD63 that is associated with LE/MVB, yet no change in loci in panels B is observed for either cell line. Moreover, it is not apparent from the images that the overall CD63 intensity actually increases as shown in the graph. Can this data in C be verified by immunoblot for both cell lines?

We thank the reviewer for this comment and agree that using the term "loci" was not a good choice and was confusing. Hence, we clarified the terminology and provided full quantification of ASO, LBPA and CD63 spots for both cell lines (Fig. 1B, 1D, 2B). Next, to assess overall CD63 level in the cell (so to include the possible difference in CD63/ASO/LBPA abundance per each spot and variability in the number of spots) we also included quantification of overall signal intensity for each marker across all ASO doses (Fig. 1C, 1E, 2C). In fact upon this more detailed analysis we did not observe any changes in CD63 number of spots (Fig 1B) or intensity (1C). We identify that the reason for this discrepancy was using "CD63" channel for cell segmentation in the original study. To avoid this issue, we repeated all experiments again with PC9 and LK2 cells additionally stained with the HCS CellMask™ Blue Stain (Thermo) which significantly improved Columbus cell segmentation algorithm.

- Fig 3A and B – TEM images of the exosomes are necessary along with NTA measurements. The authors should be careful not to describe the EV population they are studying as exosomes based solely on NTA size measurement since microvesicles that bud from the plasma membrane can also be of this small size.

We thank the reviewer for this comment. We agree with this comment and we replaced term "exosomes" to more general term "extracellular vesicles" based our novel proteomic data (Figs 3E, S5, S6 and Table S1).

- Fig 3c – the authors comment in the text that PC9 cells secrete 2x more exosomes than LK2 cells, but they do not provide a statistical comparison of this in the graph in 3c.

We thank the reviewer for comment, and we conducted additional statistical analysis (Fig. 4C). Since the difference was not significant, we deleted the original claim.

- Fig 3e – the authors need to show additional markers by western. Previous work from Clotilde They published in PNAS has suggested that it is necessary to confirm the presence of CD9 and CD81 along with CD63 to state that the EVs being studied are bona fide exosomes. Other helpful markers cited in work from her lab are TSG101 and synthenin.

We thank the reviewer for this important note. We conducted thorough proteomic characterisation of the EV secreted in different conditions (Figs 3E, S5, S6 and Table S1) and detected exosome-specific (CD9, CD63, CD81 and synthenin-1) and ectosome-specific (alpha-actinin-4, basignin and F42 cell 24 antigen) markers indicating EV's heterogeneity. Hence, we use the term EV through the revised study.

- Fig 3F and G – it is really intriguing that the authors found the LK2 cells packaged more of the ASO into exosomes than the PC9 cells, which fits with the increased potency of the ASO in the PC9 cells. However, it is not clear that this increased packaging is occurring based on the data in in Figs 1 and 2. It would be expected that the LK2 cells have increased localization of the ASO to LE/MVB, but that is not clear from the data provided as a colocalization was not performed, and the LK2 cells do not have increased ASO loci, which also might be expected if it is localized and packaged into MVBs at a greater rate, compared to PC9 cells.

We thank the reviewer for this insightful comment. We conducted colocalization analysis and found no difference between PC9 and LK2 cells (Fig. 1F). This may be related to the faster ASO secretion by LK2 cells. We also agree with the reviewer that LE/MVB is a highly dynamic compartment and trafficking/secretion rates can contribute to the colocalization level. Moreover, we characterised EV composition and found that LK2 secrete 15 exclusive RNA-binding protein upon ASO treatment (Fig. S6 and Table S2). These proteins were not implicated in ASO trafficking to our best knowledge and we hope that these will be investigated/referenced in the future.

- Fig S2B – in the text the authors set out to verify if the EVs they have collected from mouse plasma are derived from the human PC9 xenograft and they conclude from the data that the EVs are not from the xenograft. However, they do not go on to further address by additional methods whether they are able to detect EVs from the xenograft tumor in the mice. Do the authors think this means that overall, systemic administration of the ASO affects EV production in cells throughout the mice as they describe for the data shown in 3H and 3I? This would still agree with some of their in vitro work suggesting that in general, the ASO can affect exosome production, but it complicates the ability to confirm this is happening in the tumor cells themselves in vivo, which is important for understanding how well the ASO can target Kras in the tumor cells. Can they also perform this study with the LK2 cells and use immunofluorescence of tumor sections to look at how much of the ASO is retained in the LK2 vs PC9 cells? Or measure the ASO in a piece of tumor, assuming there are not too many other cell types present and it is mostly tumor cell mass.

We thank the reviewer for this important comment. We agree that the observed changes in EV concentration and size as well as ASO tumour targeting requires further deeper investigation and we removed Fig. 3H and 3I from the manuscript. In the present study it was important to test if ASO can be recycled in EVs in vivo. Our data firmly supports that up to 5% ASO can be recycled in EVs (Table 1, Figs. 4D and 4E). It is extremely interesting to understand the origin of these EV but this can be conducted in the future and wasn't in the original scope of work. We attempted to identify the origin of these EV (Fig. S10B) but could not detect any human CD63 which would originate from the PC9 xenograft cells. We cannot exclude that the number of these EV is small and below the detection limits. Moreover, using resistant tumour cells such as LK2 where we do not expect to see activity would be ethically restricted so other models should be considered to understand ASO resistance in vivo and our novel markers identified in the present study can be explored in these models in the future.

- Fig 4 – It is critical to show that this U18666A inhibits exosome secretion; this data does not appear to be present. Also, it seems surprising that the ASO does not appear to accumulate in the cells when U18666A is used. If the ASO is being packaged trafficked to LE/MVB and packaged for secretion as exosomes, it would be expected that this drug would promote its accumulation in the cells since they cannot recycle it through exosomes. Can the authors explain this? Also, this drug seems to differentially affect CD63 loci/intensity/loci size between the two cell lines even in basal conditions without the ASO. Why is this? This finding raises questions about how robust this drug is and suggests another approach should be used.

We thank the reviewer for this comment. As we mentioned above, our aim was to block the bulk EV secretion by blocking late endosome trafficking with the well-established approach (REF 38 and 39). We validated this approach by observing the clustering of CD63 late endosomes (Fig 5A) and we observed similar response in both cell lines (Fig. S11A and S11B). In addition, we showed that Rab27a knockdown using Rab27A siRNA improved ASO efficacy in LK2 cells (Fig. 5E). We agree that the overall intracellular ASO level did not corresponded to the functional activity improvement but it is important to mention that we show here that the overall intracellular ASO level in the cell is not related to ASO activity (Fig. 1D and 1E, ref 18). Importantly, we link here functional ASO activity to the spatial distribution of LBPA across late endosomes population and ASO delivery to this specific compartment (Fig 2D and S1D). Therefore, we included CD63 and LBPA quantification (Fig. S11). Although U18666A can be a robust drug for blocking bulk EV secretion, we agree with the reviewer that further identification of exact EV population involved in the ASO recycling will be helpful in order to clarify the contribution of the specific EV subsets (exosomes, ectosomes, microvesicles) in the future studies.

- Fig 5 C and D – this data questions the importance of this pathway in helping LK2 cells resist the ASO because use of U18666A barely rescues the ability of AZD4785 to decrease KRas expression in LK2 cells. Again, maybe another approach targeting exosome secretion if this pathway is important can provide a better result.

We thank the reviewer for this question. Since we have not optimised KRAS protein detection method in terms of the protein half-life we excluded Fig. 5C and 5D data and focused on the KRAS mRNA expression level as a major read-out for the ASO activity.

Reviewers' comments:

Reviewer #1 (Remarks to the Author):

The authors responded to most of my comments and significantly improved the study.

-The authors may consider highlighting the fact that majority of ASO is at the EV surface, (fig S3A) as mentioned in their response, this is important for the proposed mechanism. This could be mentioned in the discussion.

-COPII established function is in the early secretory pathway (EGolgi transport). The Authors mentioned twice a role of COPII related to endosome. I still believe that this isolated observation does not bring strength to this study and may be detrimental to the paper, when readers have a background in membrane trafficking.

Reviewer #2 (Remarks to the Author):

A major issue with the original manuscript was better characterization of EVs and proof of EV involvement in their model. Thank you to the authors for including additional analyses and approaches to address these problems. However, a couple of issues still remain here. Namely, the impact of recycling via EVs on good productive vs poor productive uptake is not clear due to a lack of sufficient and convincing functional validation of EV-mediated secretion in regulating ASO efficacy. Also, the role of LBPA remains unclear and confusing.

Technically, for Rab27a siRNA, confirmation of decreased exosome secretion is needed. This analysis may include NTA measurements and immunoblots of markers based on normalization to cell number. A similar confirmation of decreased exosome secretion should be shown for U18666A; this data does not appear to present for their particular experiments, although they referenced prior literature to justify use of this approach. Additionally, an immunoblot for knockdown of Rab27a and the primary data (e.g. representative image) from which the quantification in S11 C and D are generated is necessary to include.

Conceptually, the effect with the Rab27a siRNA does not appear so striking. Given that EV-mediated recycling is a central conclusion of their paper, are there other readouts that can be performed or other perturbations employed to better test this part of their model? As the authors mention in their rebuttal, inhibiting EV recycling alone may not be enough, so why not also increase LBPA levels, as well, as they suggest? Thus, these data seem to hint that that EV recycling is not a key player in ASO resistance as the authors are concluding. For example, Rab27a depletion may simply leave the ASO trapped in LEs, but without LBPA, it cannot be released into cells; this trapping alone could be sufficient to confer "poor productive uptake." Also, these LEs with trapped ASO could traffic to the lysosome, eliminating the need for EV-recycling, but still allowing for ASO resistance. The authors do cite prior literature demonstrating a role for the lysosome in their conclusion, but it is not clear if their own study here is truly demonstrating an alternative fate for the ASOs via EV secretion as opposed to lysosomal degradation.

While it is an interesting hypothesis that both LBPA and EV recycling are involved, a conceptual issue here is that LBPA has been linked to exosome production through its ability to recruit Alix to LE, which in turn recruits ESCRTIII components to the LE. Recently published work from Jean Gruenberg's group in the Journal of Cell Biology 2020 detailed this mechanism. Based on these findings, it might be expected that LBPA may actually promote EV-mediated recycling. Therefore, is there a way to demonstrate LBPA-mediated backfusion, which is proposed to facilitate "good productive uptake" in this manuscript? This would be critical to prove the role of LBPA as one of the key limiting factors in ASO activity, another central aspect of their study.

Reviewers' comments:

Reviewer #1 (Remarks to the Author):

The authors responded to most of my comments and significantly improved the study.

We are grateful to reviewer for this appreciating comment.

-The authors may consider highlighting the fact that majority of ASO is at the EV surface, (fig S3A) as mentioned in their response, this is important for the proposed mechanism. This could be mentioned in the discussion.

We agree that the ASO localisation on EV surface supports ASO recycling from the LE lumen and included following text in the discussion:

Page 9, Ln 19, . "We found that AZD4785 is recycled on the external EV surface, suggesting that PS-ASO can either be recycled directly from the LE lumen via LE..."

-COPII established function is in the early secretory pathway (EGolgi transport). The Authors mentioned twice a role of COPII related to endosome. I still believe that this isolated observation does not bring strength to this study and may be detrimental to the paper, when readers have a background in membrane trafficking.

We agree with the reviewer that the major coat protein complex (COP) II function is ER-Golgi vesicular traffic. Hence, we deleted the references to COPII pathway:

Page 3, Ln12. "During the final phase, PS-ASO escapes from late endosomes (LE) to the cytosol with assistance of lysobisphosphatidic acid (LBPA), annexin A2 and the recruitment of COPII vesicles and annexin A2^{18, 19}."

Page 9, Ln 47. "Modulation of intracellular PS-ASO trafficking by using small molecules regulating Alix or LBPA^{22, 20} or targeting Rab27A⁴⁰, SMPD3⁴² or novel Alix-dependent²² EV biogenesis pathways such as Rab27a or SMPD3 or even COPII^{40, 41} are novel strategies overcoming non-productive uptake in select tumour cells and further analysis of EV role in ASO productive uptake across a varied panel of models and/or diseases are required in the future."

Reviewer #2 (Remarks to the Author):

A major issue with the original manuscript was better characterization of EVs and proof of EV involvement in their model. Thank you to the authors for including additional analyses and approaches to address these problems. However, a couple of issues still remain here. Namely, the impact of recycling via EVs on good productive vs poor productive uptake is not clear due to a lack of sufficient and convincing functional validation of EV-mediated secretion in regulating ASO efficacy. Also, the role of LBPA remains unclear and confusing.

We thank the reviewer for appreciating novel included data and insightful comments. We found that modulating EV recycling with the inhibitors of Rab27A and SMPD3 pathway prevented ASO clearance and improved ASO potency 2-5.5 fold times in LK2 cells with poor productive uptake. However, EV recycling clearance mechanism is not significant in PC9 cells with good productive uptake and we found that this is predominantly influenced by early ASO engagement with LBPA in LE. In fact, stimulation of LBPA in poor productive cells LK2 with U18666A and thioperamide maleate improved AZD4785 potency 56-147 fold times. We believe that these data altogether (summarised

in Fig S13) highlight ASO clearance via EVs is less significant in ASO efficacy, and thus pave the way for future studies to be focused on LBPA spatial distribution as well as LBPA-dependent back-fusion mechanism having a more pronounced effect on ASO endosomal escape. This will also be critical for the translation ASO applications to treat various diseases which were previously deemed “undruggable”:

Page 7, Ln 27. “Intraluminal vesicles formed in LE compartment can also be secreted as exosomes¹⁹ and to investigate the role of exosome-specific clearance, we knockdown Rab27a, a well-established regulator of exosome secretion in tumour cells⁴⁰ and used small inhibitor of sphingomyelin phosphodiesterase 3 pathway, 3-O-Methyl-sphingomyelin⁴¹. Rab27a knockdown by using siRNA resulted in the reduction of Rab27A to $\approx 15\%$ level in both cell lines (Fig. S12E). Importantly, Rab27A knockdown and SMPD3 inhibition reduced the secretion of CD63+/CD81+ EV by LK2 cells and improved AZD4785 productive uptake ≈ 2 and ≈ 5.5 fold, correspondingly (Table 2, Fig. 5E and Fig S12H). Rab27a knockdown in PC9 cells had no effect on CD63+/CD81+ secretion as well as AZD4785 efficacy in PC9 cells (Table 2, Fig. 5D, Fig S12I). Moreover, inhibition of sphingomyelin phosphodiesterase 3 pathway in PC9/LK2 cells reduced the secretion of CD63+/CD81+ EVs but had no effect on AZD4785 efficacy ≈ 2 fold (Fig. S12C and S12G)”.

Page 9, Ln42. “Altogether, these data strongly indicate mainly LBPA engagement in LE and to a less extent ASO clearance via the EV recycling pathway influence the productive ASO uptake in LK2 cells (Fig. S13). Importantly, rapid AZD4785 delivery to LBPA+ LE in PC9 cells and LBPA engagement dominates over EV recycling clearance pathway enabling good productive uptake hence future studies should be focused on the further understanding of the LBPA spatial distribution as well as the LBPA-mediated “back-fusion” mechanism of endosomal escape (Fig. S13). Modulation of intracellular PS-ASO trafficking by using small molecules regulating Alix-or LBPA²⁵⁻³⁰ or targeting Rab27A⁴⁰, SMPD3⁴¹ or novel Alix-dependent³⁷ EV biogenesis pathways such as Rab27a-or SAAPD3-or even GORL^{40, 41} are novel strategies overcoming non-productive uptake in select tumour cells and further analysis of EV role in ASO productive uptake across a varied panel of models and/or diseases are required in the future.”

Technically, for Rab27a siRNA, confirmation of decreased exosome secretion is needed. This analysis may include NTA measurements and immunoblots of markers based on normalization to cell number. A similar confirmation of decreased exosome secretion should be shown for U18666A; this data does not appear to present for their particular experiments, although they referenced prior literature to justify use of this approach. Additionally, an immunoblot for knockdown of Rab27a and the primary data (e.g. representative image) from which the quantification in S11 C and D are generated is necessary to include.

We agree with the reviewer that Rab27A pathway might be specific for the particular cancer cell lines so we applied original quantitative CD63 capture bead assay developed by Matias Ostrowski and Clotilde They which was used to discover the role of Rab27A in exosome secretion (Ostrowski et al., Nat Cell Bio, 2010). In this assay EVs are captured from the cell media by CD63 capture beads and quantified by staining for CD81. We addressed this comment by including novel data:

1. Rab27A expression levels reduced to 15% after siRNA knockdown in both cell lines:

Page 7, Ln 30 “Rab27A knockdown by using siRNA resulted in the reduction of Rab27A to $\approx 15\%$ level in both cell lines (Fig. S12E).”

2. We measured the effect of Rab27A knockdown onto CD63+/CD81+ EV secretion and found that it blocks EV secretion by LK2 cells but had no effect onto PC9 cells. To validate the role of EV recycling in PC9 cells we included additional inhibitor of alternative, sphingomyelin phosphodiesterase 3-dependent EV biogenesis pathway. Again, this inhibitor inhibited EV secretion by PC9 cells but had no effects onto ASO potency:

Page 7, Ln27. "Intraluminal vesicles formed in LE compartment can also be secreted as exosomes¹⁹ and to investigate the role of exosome-specific clearance, we knockdown Rab27a, a well-established regulator of exosome secretion in tumour cells⁴⁰ and used small inhibitor of sphingomyelin phosphodiesterase 3 pathway, 3-O-Methyl-sphingomyelin⁴¹. Rab27A knockdown by using siRNA resulted in the reduction of Rab27A to ~15% level in both cell lines (Fig. S12E). Importantly, Rab27A knockdown and SMPD3 inhibition reduced the secretion of CD63+/CD81+ EV by LK2 cells and improved AZD4785 productive uptake ~2 and ~5.5 fold, correspondingly (Table 2, Fig. 5E and Fig S12H). Rab27a knockdown in PC9 cells had no effect on CD63+/CD81+ secretion as well as AZD4785 efficacy in PC9 cells (Table 2, Fig. 5D, Fig S12H). Moreover, inhibition of sphingomyelin phosphodiesterase 3 pathway in PC9/LK2 cells reduced the secretion of CD63+/CD81+ EVs but had no effect on AZD4785 efficacy ~2-4 fold (Fig. S12C and S12G)."

Page 9, Ln34. "Interestingly, in PC9 cells secretion of CD63+/CD81+ EVs was Rab27A independent and inhibition of EV release with sphingomyelin phosphodiesterase 3 inhibitor did not influence AZD4785 potency. However, in LK2 cells Rab27A knockdown and inhibition of sphingomyelin phosphodiesterase 3 reduced CD63+/CD81+ EV secretion and improved AZD4785 potency but only by ~2 and ~5.5 fold, correspondingly. In LK2 cells, U18666A treatment causes multiple effects including accumulation of LBPA so next we tested the effects of thioperamide maleate which specifically increases LBPA levels in LE⁴² and it improved AZD4785 activity in LK2 cells 56-fold. Altogether these data strongly indicate mainly LBPA engagement in LE and to a less extent ASO clearance via the EV recycling pathway influence the productive ASO uptake in LK2 cells (Fig. S13)."

3. Interestingly, addition of U18666A stimulate secretion of CD63+/CD81 by both cell lines. To our best data this is very experimental evidence that U18666A stimulates LBPA accumulation engagement in LE (see below) and acts primarily via LBPA/ASO engagement mechanism enabling endosomal escape:

Page 7, Ln11 "Surprisingly, U18666A induced secretion of CD63+/CD81+ EVs as detected by CD63 capture bead assay (Fig. S12A and S12B)."

Page 9, Ln 28 "To find out the exact contribution of the EV recycling pathway to ASO functional activity, we attempted to inhibit EV secretion pathway by blocking endosome maturation with U18666A^{37, 38} as well as by modulating of Rab27A-dependent⁴⁰ and sphingomyelin phosphodiesterase 3-dependent⁴¹ exosome biogenesis pathways. Interestingly, in agreement with previous reports U18666A treatment induced CD63 accumulation. Unexpectedly, it stimulated secretion of CD63+/CD81+ EVs by both cell lines, most likely by inducing LBPA accumulation in LE and activation of recently-established novel LBPA/Alix EV secretion pathway^{27, 43}."

4. Representative images showing LBPA stimulation by U18666A and thioperamide maleate which were used for the quantification of the graphs S11 C and S11D are included in the supplementary materials (Fig.S11C, and S11E).

Conceptually, the effect with the Rab27a siRNA does not appear so striking. Given that EV-mediated recycling is a central conclusion of their paper, are there other readouts that can be performed or other perturbations employed to better test this part of their model? As the authors mention in

their rebuttal, inhibiting EV recycling alone may not be enough, so why not also increase LBPA levels, as well, as they suggest? Thus, these data seem to hint that that EV recycling is not a key player in ASO resistance as the authors are concluding. For example, Rab27a depletion may simply leave the ASO trapped in LEs, but without LBPA, it cannot be released into cells; this trapping alone could be sufficient to confer “poor productive uptake.” Also, these LEs with trapped ASO could traffic to the lysosome, eliminating the need for EV-recycling, but still allowing for ASO resistance. The authors do cite prior literature demonstrating a role for the lysosome in their conclusion, but it is not clear if their own study here is truly demonstrating an alternative fate for the ASOs via EV secretion as opposed to lysosomal degradation.

We agree with the reviewer that LBPA/ASO engagement is the key mechanism defining poor vs good productive uptake in tumour cells and we achieved significant ASO activity (up to 147 fold improvement) in poor productive cells by stimulating LBPA accumulation with U18666A and thioperamide maleate. However, inhibiting EV recycling in poor productive uptake cells LK2 also improved ASO functional activity though at much lesser scale (2-5.5 fold). We stressed the superiority of the LBPA engagement mechanism in the discussion:

Page 9, Ln 41. Altogether, these data strongly indicate mainly LBPA engagement in LE and to a less extent ASO clearance via the EV recycling pathway influence the productive ASO uptake in LK2 cells (Fig. S13). Importantly, rapid AZD4785 delivery to LBPA+ LE in PC9 cells and LBPA engagement dominates over EV recycling clearance pathway enabling good productive uptake hence future studies should be focused on the further understanding of the LBPA spatial distribution as well as the LBPA-mediated “back-fusion” mechanism of endosomal escape (Fig. S13). Modulation of intracellular PS-ASO trafficking by using small molecules regulating Alix or LBPA^{35, 36} or targeting Rab27A⁴⁰, SMPD3⁴¹ or novel Alix-dependent³⁷ EV biogenesis pathways such as Rab27a or SMPD3 or even GPR^{40, 41} are novel strategies overcoming non-productive uptake in select tumour cells and further analysis of EV role in ASO productive uptake across a varied panel of models and/or diseases are required in the future.

While it is an interesting hypothesis that both LBPA and EV recycling are involved, a conceptual issue here is that LBPA has been linked to exosome production through its ability to recruit Alix to LE, which in turn recruits ESCRTIII components to the LE. Recently published work from Jean Gruenberg’s group in the Journal of Cell Biology 2020 detailed this mechanism. Based on these findings, it might be expected that LBPA may actually promote EV-mediated recycling. Therefore, is there a way to demonstrate LBPA-mediated backfusion, which is proposed to facilitate “good productive uptake” in this manuscript? This would be critical to prove the role of LBPA as one of the key limiting factors in ASO activity, another central aspect of their study.

We thank the reviewer for highlighting Larious et al., 2020 paper showing the novel LBPA/Alix mechanism of ESCRTIII recruitment and secretion of CD63/CD81 exosomes. This is highly relevant to our study since accumulation of LBPA could drive the balance between the ASO endosomal escape and ASO clearance via EV pathway. Our experimental evidence clearly indicate that LBPA/ASO engagement in LE dominates over ASO EV recycling clearance mechanism:

1. PC9 cells show high LBPA/ASO engagement level at the baseline (Figs. 2A, 2D, S1C) and modulation of EV secretion (stimulation with U18666A (Figs S12B) or inhibition with 3-O-Methylsphingomyelin (Fig S12C) had no effect on ASO efficacy.
2. LK2 cells show poor LBPA/ASO engagement at the baseline (Figs. 2A, 2D, S1C) and stimulation of LBPA accumulation in LE (Figs. S11E and S11F) with thioperamide maleate and U18666A improves

ASO potency ≈ 56 and ≈ 147 fold times, respectively. In fact, we observed stimulation of CD63+/CD81+ EVs secretion upon LBPA stimulation with U18666A (Fig. S12B) and thioperamide maleate (Fig S12D). However, inhibition of ASO clearance via EVs by using Rab27A siRNA (Fig S12F) and SMPD3 inhibitor (Fig. S12D) improved ASO potency only modestly (2 and 5.5. fold times, respectively) and an increase in LBPA-dependent EV secretion upon U18666A or thioperamide maleate treatment was unlikely to be sufficient to counteract LBPA/ASO engagement in LE. However, we agree with the reviewer that the contribution of various EV recycling pathways should be assessed in the future and we included citation for Lariou et al., 2020 (ref 52) and stressed the superiority of the LBPA/ASO engagement mechanism in the discussion:

Page 9, Ln 28 “To find out the exact contribution of the EV recycling pathway to ASO functional activity, we attempted to inhibit EV secretion pathway by blocking endosome maturation with U18666A^{37,38} as well as by modulating of Rab27A-dependent⁴⁰ and sphingomyelin phosphodiesterase 3-dependent⁴¹ exosome biogenesis pathways. Interestingly, in agreement with previous reports U18666A treatment induced CD63 accumulation. Unexpectedly, it stimulated secretion of CD63+/CD81+ EVs by both cell lines, most likely by inducing LBPA accumulation in LE and activation of recently-established novel LBPA/Alix EV secretion pathway²². Interestingly, in PC9 cells secretion of CD63+/CD81+ EVs was Rab27A independent and inhibition of EV release with sphingomyelin phosphodiesterase 3 inhibitor did not influence AZD4785 potency. However, in LK2 cells Rab27A knockdown and inhibition of sphingomyelin phosphodiesterase 3 reduced CD63+/CD81+ EV secretion and improved AZD4785 potency but only by ≈ 2 and ≈ 5.5 fold correspondingly. In LK2 cells, U18666A treatment causes multiple effects including accumulation of LBPA so next we tested the effects of thioperamide maleate which specifically increases LBPA levels in LE²⁶ and it improved AZD4785 activity in LK2 cells 56-fold. Altogether, these data strongly indicate mainly LBPA engagement in LE and to a less extent ASO clearance via the EV recycling pathway influence the productive ASO uptake in LK2 cells (Fig. S13). Importantly, rapid AZD4785 delivery to LBPA+ LE in PC9 cells and LBPA engagement dominates over EV recycling clearance pathway enabling good productive uptake hence future studies should be focused on the further understanding of the LBPA spatial distribution as well as the LBPA-mediated “back-fusion” mechanism of endosomal escape (Fig. S13). Modulation of intracellular PS-ASO trafficking by using small molecules regulating Alix or LBPA^{23, 39} or targeting Rab27A⁴⁰, SMPD3⁴² or novel Alix-dependent²² EV biogenesis pathways such as Rab27a or SMPD3 or even COP4^{40, 41} are novel strategies overcoming non-productive uptake in select tumour cells and further analysis of EV role in ASO productive uptake across a varied panel of models and/or diseases are required in the future.”

REVIEWERS' COMMENTS:

Reviewer #1 (Remarks to the Author):

the revised version is greatly improved. the authors addressed all the points raised by the reviewers. Congratulations.

Reviewers' comments:

REVIEWERS' COMMENTS:

Reviewer #1 (Remarks to the Author):

the revised version is greatly improved. the authors addressed all the points raised by the reviewers. Congratulations.

We are very grateful to reviewers and editors for the comments and suggestions which guided the study and helped to find out the mechanism of antisense oligonucleotide's productive uptake. This is an important step enabling oligonucleotide's target selection and application in cancer and other therapeutic areas.